# Identifying and characterizing pesticide use on 9,000 fields of organic agriculture

Ashley E. Larsen [1]✉, L. Claire Powers [1,2] & Sofie McComb [1,3]

Notwithstanding popular perception, the environmental impacts of organic agriculture, particularly with respect to pesticide use, are not well established. Fueling the impasse is the general lack of data on comparable organic and conventional agricultural fields. We identify the location of ~9,000 organic fields from 2013 to 2019 using field-level crop and pesticide use data, along with state certification data, for Kern County, CA, one of the US' most valuable crop producing counties. We parse apart how being organic relative to conventional affects decisions to spray pesticides and, if spraying, how much to spray using both raw and yield gap-adjusted pesticide application rates, based on a global meta-analysis. We show the expected probability of spraying any pesticides is reduced by about 30 percentage points for organic relative to conventional fields, across different metrics of pesticide use including overall weight applied and coarse ecotoxicity metrics. We report little difference, on average, in pesticide use for organic and conventional fields that do spray, though observe substantial crop-specific heterogeneity.

[1] Bren School of Environmental Science & Management, UC, Santa Barbara, CA, USA. [2] Environmental Studies Program, University of Colorado Boulder, Boulder, CO, USA. [3] Faculty of Forestry, University of British Columbia, Vancouver, Canada. ✉email: larsen@bren.ucsb.edu

Agriculture covers about 40% of arable land globally[1] and is a leading cause of environmental degradation[1]. Despite tremendous gains in agricultural production in the past several decades, hunger and malnutrition remain a challenge[2], and demand for agricultural products continues to increase[3]. Finding scalable ways to improve the sustainability of agricultural production is critical to supporting a growing population and mitigating damage to human and environmental health.

Organic agriculture is a commonly suggested approach to improving the sustainability of agricultural production. Organic agriculture currently covers only ~1.5% of global agricultural land but is growing rapidly in extent and sales[4,5]. For example, between 2000 and 2015, global organic agriculture grew from 15 million ha to 51 million ha[2] and has since exceeded 73 million ha[5]. The rapid increase in consumer demand for organics is driven by a variety of factors mostly related to nutrition, food and farmworker safety, and environmental concerns[4]. Similar concerns have prompted numerous policy initiatives to promote organic production, such as the European Union's Farm to Fork Strategy[6,7]. Despite popular perception, understanding the benefits and drawbacks of organic agriculture on a per output basis remains an area of active research[8–13].

The crux of the debate, from the environmental sustainability perspective, is whether the reduction in negative ecological and environmental impacts on-field compensates for the reduction in yields[14,15] and increased yield variability[16] that has been observed for most organically produced crops in actual field surveys[8,16,17]. Though seemingly simple, addressing this question for even a subset of environmental outcomes is plagued by methodological challenges stemming from a lack of a valid comparison group[18]. Organic fields are unlikely to be randomly placed on the landscape, nor are organic farmers likely a random draw of the broader farming community. In other words, organic farms might grow a systematically different suite of crops on systematically better or worse soil, or be produced by farmers with systematically different environmental or health behaviors than their conventional neighbors. Though such challenges can and have been addressed through long-term field trials[19,20], understanding the difference between organic and conventional fields in real-world settings is crucial for understanding the merits of organic production practices at scale. However, surveys of yields and practices across a small number of willing farmers will likely be plagued by selection bias. This sample-selection challenge is further compounded by a paucity of field-level data on inputs and/or outputs in general, and on organic fields in particular[11], which complicates comparisons.

While only one metric of sustainability, pesticides are highly salient to consumers[11,21]. Historically, widespread use of persistent, broad-spectrum, and bioaccumulating chemicals such as organochlorines and organophosphates had severe negative impacts on humans, other mammals, and birds[22,23]; impacts which helped spur the early organic movement[24,25]. As those risks were increasingly recognized, a new generation of chemicals was developed with a particular focus on reducing human-health risks. While the development and uptake of these pesticides have limited direct mammal and bird mortality in recent decades[26], many remain highly toxic to other organisms[27–29]. Further, population-level ecological effects through food web interactions or sub-lethal impacts (e.g., behavioral changes and reduced migratory navigation) remain a concern for higher taxa[30,31].

Organic agriculture is commonly perceived to be chemical-free, though organic as a regulatory definition, at least in the US, generally restricts the type of inputs applied rather than the amount[21]. The regulation does not itself require chemical-free farming and organic compliance does not always imply low toxicity to ecological or environmental endpoints. For example,

organic-acceptable active ingredients such as copper[21], pyrethrin, and azadirachtin are toxic to aquatic organisms[32,33]. Furthermore, since pesticide residue testing often focuses on synthetic chemicals of high human toxicity[34,35] and field-level data on production or certification are rarely available, little is known about pesticide use practices on organic fields.

Our goal is to quantify the differences in total pesticide use and pesticides of specific concern to different ecological and environmental endpoints to further understanding of the environmental benefits and drawbacks of different production systems. We harmonize and aggregate several data sources to identify the spatial location of organic crop fields and rely on unique, field-level crop and pesticide use data from Kern County, California to understand pesticide use differences. Kern County produces a variety of high-value fruit and vegetable crops and is consistently one of California's and US' most valuable crop-producing counties by sales[36]. Due to the number of different products and chemicals applied in our study area, we rely on pesticide use and coarse metrics of ecotoxicity based on the pesticide product label, which reflects the exceedance of toxicity thresholds to different endpoints. We acknowledge pesticide use may not always reflect potential environmental harm[37,38], and even thresholds of toxicity cannot differentiate between the hazards posed by two chemicals that both exceed regulatory thresholds, but that differ in toxicity. We consider and discuss toxicity metrics and limitations throughout.

Our investigation primarily relies on double hurdle models to parse apart the decision to spray pesticides from the decision of how much to spray. Using these models, we evaluate (1) overall differences between organic and conventional fields with respect to the decisions to spray and how much to spray for total pesticide use and pesticides of potential hazard to a range of different endpoints, (2) crop-specific differences in pesticide use decisions between organic and conventional fields for five crops commonly grown with both organic and conventional practices, and (3) how adjusting for yield gaps may influence the overall results. Our results suggest organic agriculture is more likely to be "pesticide-free", but organic fields that are sprayed tend to receive similar levels of pesticides as their conventional neighbors.

## Results

**Summary statistics.** Our sample consisted of 99,533 fields, which were all permitted fields in Kern County between 2013 and 2019. Across the sample, the average field size was about 31 ha and soil quality—measured as the California Revised Storie Index[39] with a range from 1 (highest quality) to 6 (lowest quality)—averaged 1.8. Annually, about 25 kg ha$^{-1}$ of pesticide active ingredients (AI) and 45 kg ha$^{-1}$ of pesticide products (AI + adjuvants) were used on the average field. There were 1293 farms in the average year with an average farm size of 451 ha. All of these variables varied spatially throughout Kern (Fig. 1).

We identified about 9100 organic fields between 2013 and 2019 by joining pesticide use data and data on the spatial location of certified organic fields obtained by request from the California Department of Food and Agriculture (see methods). Per year, the number of organic fields ranged from a low of 936 in 2013 to a high of 1544 in 2017. In a given year between 2013 and 2019, there were about 14,200 permitted fields, and thus organic fields represented ~7–11% of fields in Kern County. On average across the entire sample, organic fields were about 44% smaller than conventional fields (Table 1); however, farms containing both conventional fields and at least one organic field in the study period were ~4 times larger than purely conventional farms (1240 ha compared to 319 ha). Organic fields also tended to be on better soil, both on average (Table 1) and when accounting for

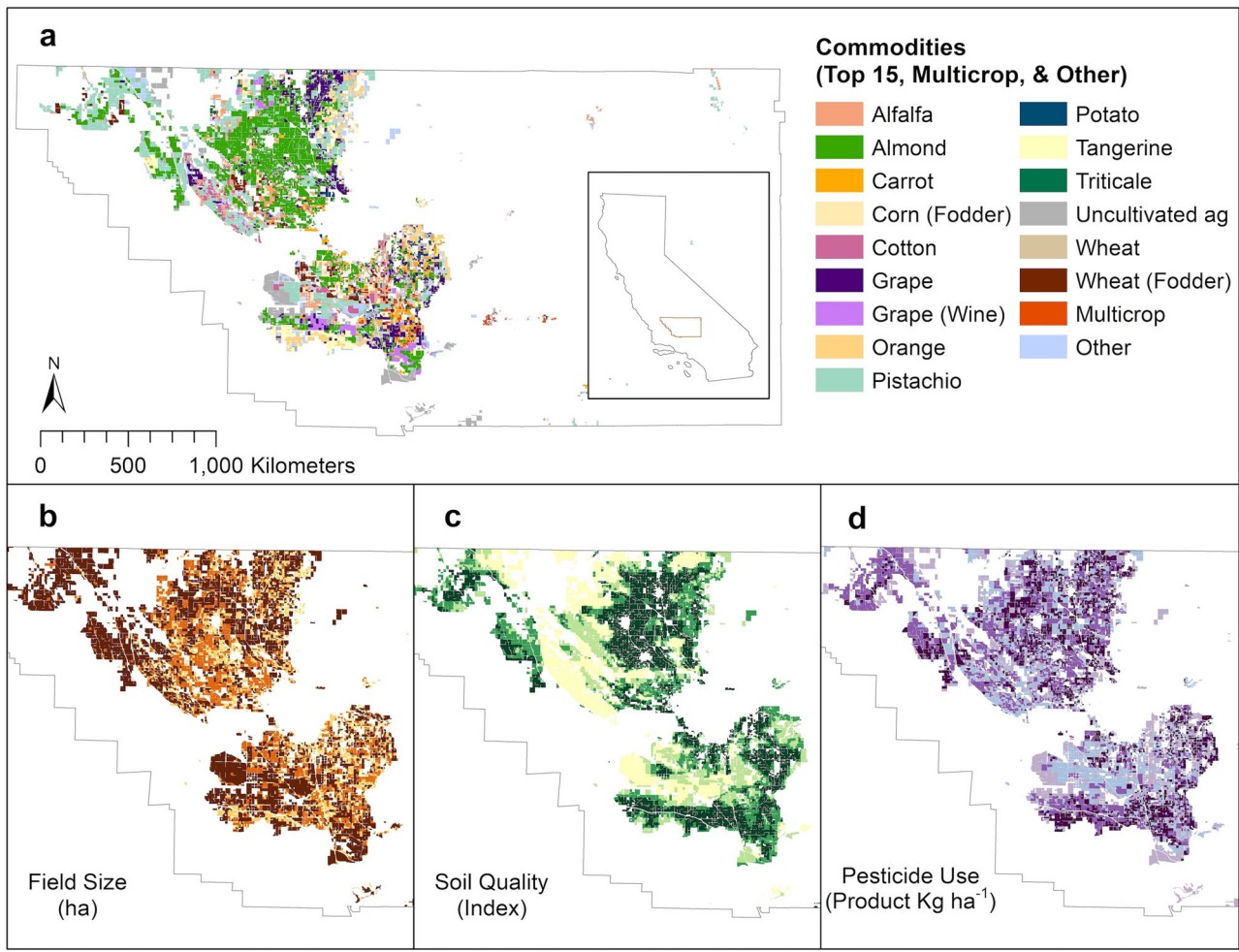

**Fig. 1 Kern Fields in 2019. a** Top 15 crops (by a number of fields) grown with all other crops grouped as "Other" and all multi-crop fields marked as "Multicrop". **b** Field size in hectares, with quartiles from 0 (light yellow) to 296 (dark red). **c** Soil quality by Storie Index, with quartiles from 1 (dark green-productive soil) to 6 (light yellow-unproductive soil). **d** Pesticide uses in kg ha$^{-1}$ of a pesticide product, with quartiles from 0 (light purple) to 2188 (dark purple). For fields with multiple values (multi-crop, active vs. inactive permits, etc.), maps illustrate the maximum value of each variable, and for crops, the most frequent crop grown.

**Table 1 Summary statistics for organic and conventional fields.**

|  |  | Total fields | Mean-field size (ha) | Mean soil quality (1 = high, 6 = low) | Mean active ingredients applied (kg ha$^{-1}$) | Mean pesticide product applied (kg ha$^{-1}$) |
|---|---|---|---|---|---|---|
| All Fields | Conv | 90,439 | 32.03 ± 0.11 | 1.87 ± <0.01 | 26.91 ± 0.27 | 47.62 ± 0.43 |
|  | Org | 9,094 | 17.91 ± 0.17 | 1.37 ± <0.01 | 8.04 ± 0.27 | 17.10 ± 0.48 |
| Carrots | Conv | 2,886 | 26.88 ± 0.25 | 1.35 ± 0.01 | 127.90 ± 3.61 | 253.46 ± 7.28 |
|  | Org | 1,403 | 29.22 ± 0.36 | 1.29 ± 0.01 | 11.67 ± 0.60 | 19.90 ± 0.98 |
| Grapes | Conv | 8443 | 27.88 ± 0.34 | 1.54 ± 0.01 | 59.83 ± 1.96 | 99.91 ± 2.51 |
|  | Org | 317 | 27.73 ± 1.37 | 1.52 ± 0.05 | 65.38 ± 4.16 | 124.91 ± 7.55 |
| Onions | Conv | 985 | 22.05 ± 0.61 | 1.59 ± 0.03 | 21.04 ± 2.69 | 39.90 ± 4.98 |
|  | Org | 141 | 23.63 ± 1.20 | 1.21 ± 0.03 | 26.65 ± 4.39 | 38.46 ± 5.28 |
| Oranges | Conv | 4,520 | 24.05 ± 0.46 | 1.43 ± 0.01 | 50.06 ± 0.89 | 92.05 ± 1.33 |
|  | Org | 141 | 26.04 ± 1.84 | 1.33 ± 0.05 | 33.16 ± 4.54 | 44.82 ± 5.13 |
| Potatoes | Conv | 2,314 | 29.07 ± 0.37 | 1.45 ± 0.01 | 46.80 ± 1.96 | 94.13 ± 4.25 |
|  | Org | 489 | 28.09 ± 0.65 | 1.30 ± 0.02 | 1.61 ± 0.17 | 4.61 ± 0.49 |

Mean ± standard error (sd/√n) for field size, soil quality, and pesticide use for all fields and for five crops commonly grown as both organic ("Org") and conventional ("Conv").

crop-specific heterogeneity (Supplementary Table 1). Top actively cultivated crops by area included carrot, potato, lettuce leaf, and tomato for organic fields and almond, pistachio, grape, and alfalfa for conventional fields, respectively.

**Lognormal hurdle models.** We began with pooled and panel data models to understand the influence of different model specification decisions (Methods, Supplementary Notes, Supplementary Tables 2 and 3). However, the mechanisms determining zero values for pesticide use may differ from those determining the level of use, and basic summary statistics suggest differences in the frequency of zero values between organic and conventional fields (Supplementary Table 4). As such, we investigate and primarily discuss double hurdle models to parse apart the decision to spray from the decision of how much to spray. For the first hurdle, we are interpreting the use of zero pesticides as the true choice of the farmer and are predicting the probability the farmer (of a given field) is "zero type" as a function of being organic or not. We do so using a random-effects probit model with covariates for field size, farm size, and soil quality, with random intercepts for farm-by-crop family and with cluster robust standard errors clustered at the farm-by-crop family (Methods, Supplementary Notes). We report the average marginal effects from the probit models. For all active ingredients, we find being organic leads to an average ~31 percentage point ($0.31 \pm 0.03$) reduction in the probability of pesticide use (Fig. 2, Supplementary Table 5). Since the weight of active ingredients does not necessarily reflect environmental harm[37], we test several other pesticide outcomes based on information on the pesticide label. The pesticide label, which is governed by the EPA[40], includes hazards statements that reflect whether one or more chemicals in

the product exceeds thresholds regarding acute toxicity to humans, as well as for different environmental and ecological outcomes. These statements are generally based on acute toxicity studies for humans, and for birds, fish, invertebrates, pollinating insects, and mammals[40] (see "Methods"). In addition, labels include a statement based on the product's potential to drift or be transported in other media, and information on the active ingredients from which target taxa can be derived, among other information. For these other pesticide use outcomes—pesticide products, insecticide products, products with a propensity to drift, products of potential hazard to fish and bees, and products of higher (EPA signal word 1–2) and lower (EPA signal word 3–4) acute toxicity to humans—we see being organic leads to an 18–31 percentage point ($0.18 \pm 0.02$ to $0.31 \pm 0.03$; Fig. 2, Supplementary Table 5) average reduction in the probability of spray, holding all else constant. Here, increasing field and farm size and higher soil quality lead to a significant increase in the average probability of using pesticides for most of the pesticide use outcomes evaluated.

In the second hurdle, we evaluate what drives the amount of pesticide use on fields that decide to spray. With the exception of lower acute human toxicity products, we see a generally negative effect of being organic, though the coefficient is only significant ($p < 0.05$) for higher acute human toxicity products (Fig. 2, Supplementary Table 5). We report the organic coefficients as the semi-elasticity, calculated from the second hurdle (log-level) model as $100(e^{\beta} - 1)$, and the standard error, derived using the delta-method implemented with the nlcom function in Stata. Here, a switch to organics leads to a significant $27 \pm 11$ % decline in the amount of use per hectare for higher toxicity products and a significant $28 \pm 14$% increase in lower toxicity products. For all

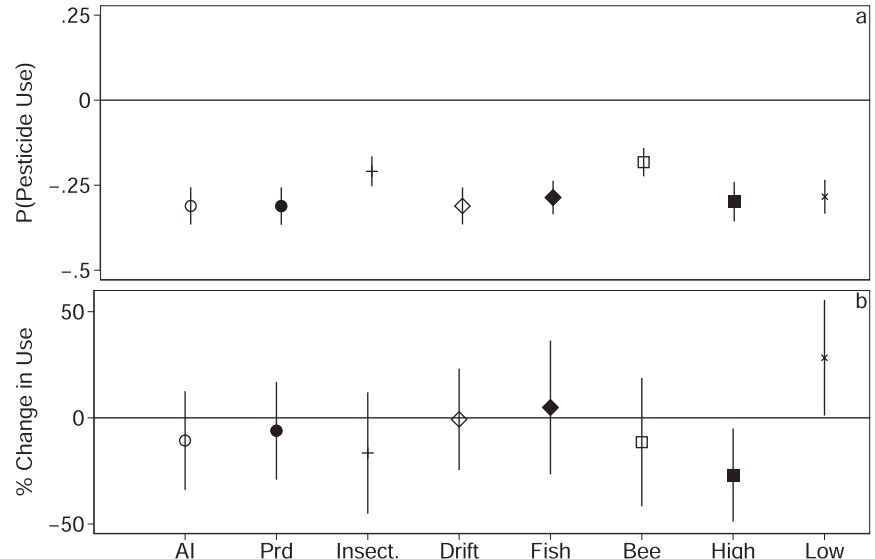

**Fig. 2 Lognormal hurdle models for different pesticide use outcomes.** Lognormal hurdle models estimating the change in the probability of pesticide use (**a**) and the percent change in pesticide use for fields with positive use (**b**) for organic relative to conventional fields. The x-axis indicates different measures of pesticide use outcomes: kg ha⁻¹ active ingredients (AI), kg ha⁻¹ products (Prd), kg ha⁻¹ of products targeting insect pests only (Insect), kg ha⁻¹ of products with a propensity to drift (Drift), kg ha⁻¹ products of potential hazard to fish and bees (Fish, Bee), as well as products of higher (EPA signal word 1–2) and lower (EPA signal word 3–4) acute human toxicity (High, Low). Across all outcomes, organic fields have a significantly lower probability of using pesticides (**a**), though there is little difference between organic and conventional fields for those that do spray, with the exception of higher and lower toxicity chemicals (**b**). Symbols indicate point estimates (mean) and error bars represent the 95% CI. All models include cluster robust standard errors clustered at the farm-by-crop family level. For the second hurdle (**b**) in Figs. 2–4, percent change is calculated from the log-level model as $100(e^{\beta} - 1)$ and standard errors are derived using the delta-method implemented with the nlcom function in Stata. All models include covariates for field size, farm size, and soil quality as well farm-by-crop family random effects. $N = 91,926$ for all specifications in the first hurdle (**a**) and $N = 68,704$ (AI), $N = 68,816$ (Prd), $N = 52,606$ (Insect.), $N = 67,988$ (Drift), $N = 60,653$ (Fish), $N = 48,254$ (Bee), $N = 61,883$ (High), and $N = 65,593$ (Low) in the second hurdle (**b**), where abbreviations are as described above. Coefficient estimates for all covariates are provided in Supplementary Table 5.

other outcomes except products of potential hazard to fish, the coefficient estimates were equivalent to a ~1–17% decrease (Fig. 2, Supplementary Table 5), though none were statistically significant.

**Crop-specific hurdle models.** Random intercepts for farm-by-crop families allow for correlation between fields shared by a farmer of a specific class of crops but force all observations to share the same overall relationship, which may hide important and policy-relevant variation in the relationship between management and pesticides for individual crops. We re-ran the log-normal hurdle model for five crops (carrot, grape, orange, potato, and onion) that were relatively widely grown both organically and conventionally (see "Methods"). This allows for unique slopes by crop type, though reduces the number of observations substantially, particularly for organics in the second hurdle. Across these five crops, we see a consistent and statistically significant 21–51 percentage point (0.21 ± 0.04 to 0.51 ± 0.03) decrease in the average probability of using any pesticide active ingredients (Fig. 3, Supplementary Table 6). The effect of organics on the second hurdle was much more variable. We see being organic significantly decreases pesticide AI on carrots by about 87 ± 3%, while significantly increasing pesticide AI on grapes by about 132 ± 33% (Fig. 3, Supplementary Table 6). Switching to organic is estimated to significantly reduce pesticide use by 64 ± 11% and 81 ± 5%, respectively, for oranges and potatoes, and result in a non-significant 54 ± 54% increase for onions (Fig. 3, Supplementary Table 6). Compared to the pooled model with all crops, these crops, on average, had a larger decrease in the probability of using any pesticides and a larger decrease in pesticide use for organic versus conventional fields that did spray (Supplementary Tables 5 and 6).

Since kg of pesticide active ingredients does not necessarily capture environmental harm, we evaluated several pesticide outcomes for grape and carrot, the two most widely grown of our top five crops. This further limited the sample size in the second

hurdle, which uses only positive observations, particularly for organic fields. Across all outcomes, a switch to organic reduced the average probability of using a given type of pesticide by about 27–51 percentage points (0.27 ± 0.05 to 0.51 ± 0.04) for carrots and 21–23 percentage points (0.21 ± 0.03 to 0.23 ± 0.3) for grapes (Supplementary Tables 7 and 8, Supplementary Fig. 1). As above, for the second hurdle, a switch to organic for carrots reduced most types of pesticide use by 80 ± 9% to 98 ± 1%, while increasing low toxicity chemicals by about 72 ± 12%. Switching to organics for grapes increased use by 126 ± 34% to 286 ± 98% (Supplementary Tables 7 and 8, Supplementary Fig. 1).

**Accounting for a yield gap.** Throughout, we have been measuring pesticide use per area rather than per unit output since we lack information on yields. This is important here since many organic crops have reduced yields relative to conventional production[14,15]. To assess the possible impact of a yield gap on our results, we multiply the use rate by the yield gap estimates in Ponisio et al.[15] for organic fields and rerun the lognormal hurdle models across all crops pooled. Doing so, of course, does not impact our coefficients in the first hurdle (decision to use any pesticides), but does shift our coefficients in the second hurdle. Across all outcomes except low toxicity chemicals, the effect of switching to organics is near zero and not significant (Fig. 4, Supplementary Table 9).

**Other sample considerations.** Observations missing crop families were dropped in any model that included family in either the random effects or the cluster robust standard errors. While 7367 fields were dropped due to missing crop families, 6684 of those were uncultivated agriculture. A small number of observations (n = 319 out of >90,000) were dropped due to missing soil quality data. Including observations with interpolated soil quality has little effect on our results. Including fields that self-reported organic increased the sample of organic fields by 407 (out of >9000), and resulted in more consistently positive, though not

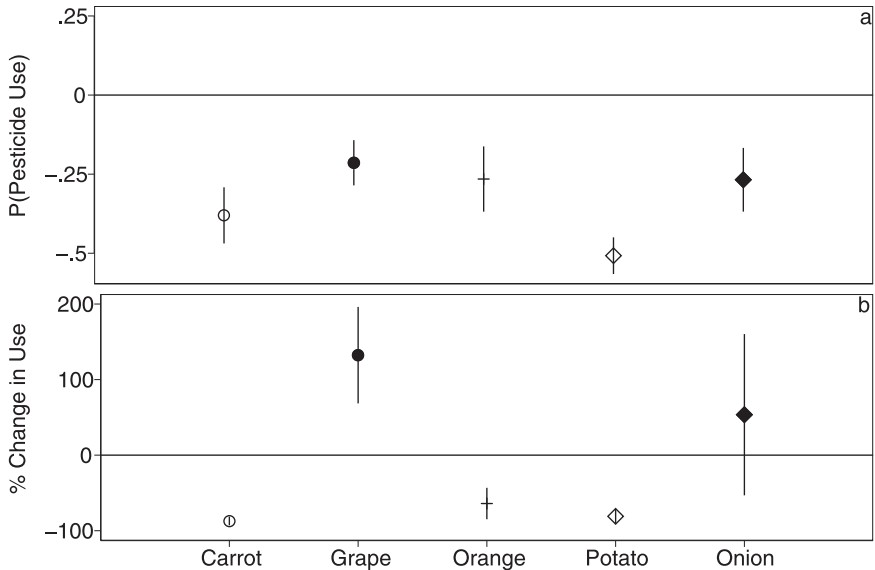

**Fig. 3 Lognormal hurdle models run separately for 5 commonly grown organic and conventional crops.** Across all five crops, organic fields have a lower probability of using any pesticide active ingredients (**a**). The effect of organic on pesticide use for fields that do spray is crop-dependent (**b**). For the second hurdle (**b**), percent change is calculated from the log-level model as $100(e^{\beta} - 1)$ and standard errors are derived using the delta-method implemented with the nlcom function in Stata. Symbols indicate point estimates (mean) and error bars represent the 95% CI. All models include heteroskedasticity robust standard errors. All models include covariates for field size, farm size, and soil quality, as well as year random intercepts. For the first hurdle, N = 4289 (Carrot), N = 8760 (Grape), N = 4654 (Orange), N = 2804 (Potato), N = 1126 (Onion). For the second hurdle, N = 2766 (Carrot), N = 7678 (Grape), N = 4316 (Orange), N = 2059 (Potato), N = 814 (Onion). Coefficient estimates for all covariates are provided in Supplementary Table 6.

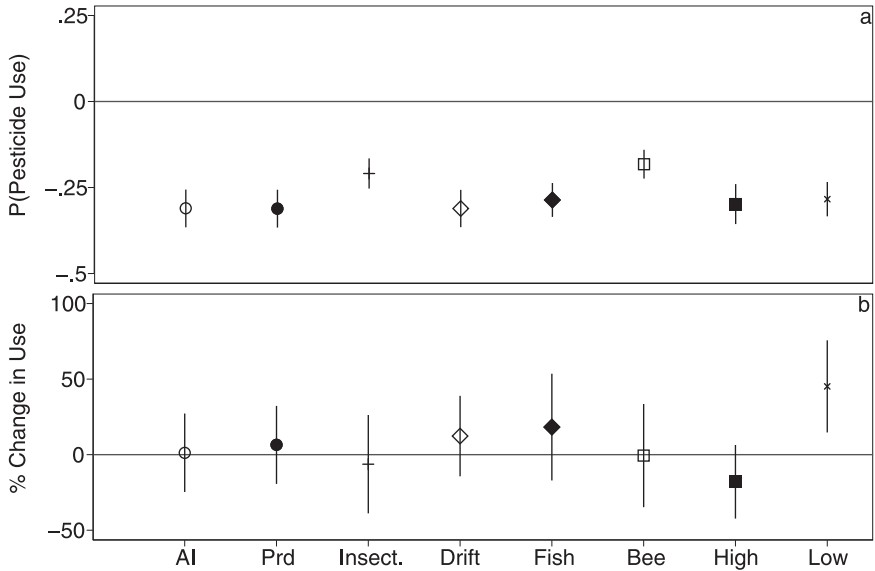

**Fig. 4 Lognormal hurdles models correcting for estimated yield gaps following Ponisio et al.[15].** Correcting for yield gaps does not affect the first hurdle (**a**), but does shift the coefficient estimates in the second hurdle up (**b**) relative to the unadjusted model (Fig. 2). Figure details are otherwise the same as Fig. 2. The x-axis indicates different pesticide use outcomes: kg ha$^{-1}$ active ingredients (AI), kg ha$^{-1}$ products (Prd), kg ha$^{-1}$ of products targeting insect pests only (Insect), kg ha$^{-1}$ of products with a propensity to drift (Drift), kg ha$^{-1}$ products of potential hazard to fish and bees (Fish, Bee), as well as products of higher (EPA signal word 1–2) and lower (EPA signal word 3–4) acute human toxicity (High, Low). Symbols indicate point estimates (mean) and error bars represent the 95% CI. All models include cluster robust standard errors clustered at the farm-by-crop family level. For the second hurdle (**b**), percent change is calculated from the log-level model as $100(e^{\beta} - 1)$ and standard errors are derived using the delta-method implemented with the nlcom function in Stata. All models include covariates for field size, farm size, and soil quality as well as farm-by-crop family random effects. $N = 91,926$ for all specifications in the first hurdle (**a**) and $N = 68,704$ (AI), $N = 68,816$ (Prd), $N = 52,606$ (Insect.), $N = 67,988$ (Drift), $N = 60,653$ (Fish), $N = 48,254$ (Bee), $N = 61,883$ (High), and $N = 65,593$ (Low) in the second hurdle (**b**). Coefficient estimates for all covariates are provided in Supplementary Table 9.

statistically significant, coefficients on organic for most pesticide outcomes in the second hurdle (Supplementary Figs. 2 and 3).

**Toxicity considerations.** As pesticide use does not always reflect potential environmental harm, we sought to investigate additional measures of toxicity as robustness tests. The data needed to calculate many toxicity metrics (e.g., pesticide load and pesticide toxicity index) were not readily available or accessible for the numerous chemicals in use and endpoints of interest. Nevertheless, to compare our binary metrics of toxicity to a continuous metric, we followed Nowell et al.[41] to calculate the Pesticide Toxicity Index for one well-studied environmental endpoint, fish. Data for other endpoints either had less coverage or were not readily available. Still, only about 70% of the chemicals in use in our study period matched the Nowell et al.[41] data or data from the Ecotox database accessed with the R package Standartox[42]. Further, organic fields were more likely to lack toxicity information. Noting the bias due to non-random missing toxicity data, we nevertheless ran our double hurdle model for fields that had products with toxicity information for, on average, 70% of the chemicals used. About a third of our organic fields that had pesticide use met this condition, while about two-thirds of the conventional fields with pesticide use met this condition. For this subsample, our highly preliminary investigation suggests a decrease in toxicity-weighted use for fish of about half for fields that are organic relative to conventional. However, a more comprehensive understanding of toxicity particularly for organic chemicals and other endpoints is needed to appropriately compare ecotoxicological outcomes between organic and conventional fields.

## Discussion

Organic agriculture is often proposed as a more sustainable and environmentally healthy alternative to mainstream, conventional approaches. Yet, a lack of data on production and inputs has hampered comparisons of conventional and organic methods[11,14,15,43] spurring ongoing debates[4,11–13,44]. Here, we sought to isolate all organic fields across one of the US' most valuable crop-producing counties and compare the effect of organic versus conventional production systems on different metrics of field-level pesticide use. Our analysis provides four main innovations: (1) for the first time, we have isolated the spatial location of thousands of organic fields using production and pesticide use data, (2) organic fields are generally smaller in size, part of larger farms, and on better soil than their conventional counterparts, (3) organic agriculture, on average, uses less pesticides than conventional production and this manifests in a lower probability of using any pesticides and similar use on fields that do spray, (4) different crop types vary considerably from the average, and in some cases, the reverse relationship is present and significant across pesticide metrics.

Field-level data from Kern County, CA illustrates around 7–11% of crop production is organically grown, based on our definition which sought to capture certified organic fields. However, these fields are not a random subset of agricultural fields in the county. Rather, there is little overlap in the most commonly grown organic and conventional crops, which was part of our motivation for including farmer-by-crop family random effects and evaluating crop-specific relationships. Organic crops differ in other ways as well. Organic crops tend to be grown on more productive soil—on average and even after controlling for crop-specific characteristics (see methods)—suggesting that any yield gap in Kern County may be an underestimate. In addition,

organic fields, while smaller in size, are generally part of farms far larger than their purely conventional counterparts, indicating the potential for selection bias in comparisons of environmental impacts. Beyond contradicting the small family farm many consumers envision, this may indicate both economy of scale benefits and a tendency for producers of organic crops to maintain a large portfolio to smooth production shocks, which may differentially affect conventional and organic crops[2]. Given the dramatic difference in farm size, future investigations into farm structure that support viable organic production in intensive production regions are warranted.

Pesticide use, though only one component of environmental or health aspects of organic production, is of paramount importance to consumers due to potential environmental and human health impacts such as water quality contamination and consumer and farm-worker wellbeing[4,45–47]. We parse apart the decision to spray pesticides from the decision of how much to spray using lognormal hurdle models. This allows for different mechanisms to influence each decision. In the first hurdle, which models the decision to use pesticides or not, the coefficient on organics, pooled across all crops, suggests a significant 18–31 percentage point reduction in the probability of spray, depending on the pesticide use metric (all active ingredients, products of potential hazard to fish, high acute human toxicity, etc.). Therefore, on average, organic fields in Kern County are more likely to be "pesticide-free", and the absence of spraying implies a lack of ecotoxicological impacts stemming from pesticide use on those fields. For the second hurdle, which models the decision of how much to spray on fields that have positive pesticide use, we see a significant 27% reduction in pesticide products (kg ha$^{-1}$) of high acute human toxicity for organic versus conventional, where high toxicity is defined as EPA acute toxicity category one or two. For other metrics of pesticide use (total active ingredients, total products, products of potential hazard to fish, etc.) where there are sufficient observations for comparison, we generally see a nonsignificant reduction.

As is often the case, the average belies high levels of crop-specific heterogeneity[48,49]. Here, we observe that variation in the organic coefficient is highly variable in the individual crop models relative to the pooled crop model, but only in the second hurdle. For crops commonly grown both organically and conventionally, we see a consistent reduction in the probability of using any pesticides, of similar magnitude as the overall average. Yet, for the second hurdle, we see some crops have very large decreases in pesticide use across metrics, and for others, particularly grapes, we see large increases across most metrics. This crop-specific heterogeneity in the effect of organic management on pesticide use is critical to acknowledge, and policies seeking to encourage changes in pesticide use behavior would be most effective if they account for individual crop relationships.

As the equalizer between low-intensive, expansive agriculture and high-intensive agriculture, yields are at the crux of most debates regarding the sustainability of different production systems[2,50]. In the absence of yield data, studies such as ours, unfortunately, rely on per area, rather than per unit production, comparisons of environmental impacts related to pesticide use and other on-farm decisions. Correcting for yield differences using published, crop-group specific yield gap estimates[15] does not, of course, change our estimated effects on the decision to spray. However, it does shift the estimated effect of organics on the amount of pesticide used across metrics such that the difference between organic and conventional is near zero, often positive, but not significant. The Ponisio et al.[15] study is an estimate based on a meta-analysis of studies done across many growing regions and could be far removed from the realities in California. Better estimates of yield differences in this region

would dramatically improve understanding of the environmental impacts of organic production.

Interestingly, we observe that field size leads to an increase in the propensity to spray across all outcomes and for most individual crops. There are both economic and ecological explanations for why increasing field size would lead to an increase in pesticide use. Larger fields provide farmers more control of their pest populations and more private benefit from their pest control actions[51]. Ecologically, larger fields may result in less spillover of natural enemies into crop fields[52,53], thus resulting in more chemical pest control[48,49,54]. Yet, with either of these explanations, we would expect field size to reduce the amount as well as the probability of treatment. Oddly, we only observe consistently significant effects of field size in the first hurdle. Though outside the scope of this study, exploring the influence of spatial arrangement of organic and conventional fields and between similar crops that may share pest or natural enemy communities may yield nuanced insight into the field size relationship observed. For farm size, we see increasing farm size leads to an increase in the propensity to treat for most outcomes, but a decrease in the amount of pesticide used on treated fields. Again, we might anticipate larger farms benefit from economies of scale, and therefore spray more because it is cheaper to do so. While our results may suggest the threshold for treatment is lower, the actual amount used on treated fields is counterintuitively inversely related to farm size.

Beyond lacking yield data, there are other caveats to our approach. For one, it is possible we have misidentified some certified organic fields. Identifying organic fields was particularly challenging for observations that provided only the Public Land Survey (PLS) Section in which the field was located. We used the pesticide use data to carefully eliminate fields using conventional pesticides, but it is possible that fields within a PLS Section that did not use conventional pesticides were still not certified organic but were misidentified as such. Second, it is also the case that not all fields following organic practices are certified organic. Small farms, in particular, may find the cost of certification prohibitive and forego it[55,56]. The inclusion of these fields in the conventional group could downward bias the coefficient on organics; however, including self-reported organic fields in a subset of years did not qualitatively affect our results. Third, while we observe permit numbers, which reflect producers, we do not observe agricultural enterprises. In other words, if a parent company has multiple labels, we fail to account for those relationships. Doing so would be important if decisions about where to plant organic fields were made at the parent company level and reflected environmental or pest conditions. Additionally, our metrics of pesticide use and potential hazards are based on binary thresholds of toxicity to different environmental endpoints. Because of the many chemicals in use, the many untested adjuvants associated, and the many environmental endpoints of interest, we were not able to apply more refined measures of toxicity[57,58]. A preliminary investigation into one well-studied endpoint hints that there may be a larger gap between organic and conventional pesticides if continuous metrics of pesticide use were more widely available. More in-depth toxicological analyses would be extremely valuable. Furthermore, though we conduct several robustness tests including different random effects and within estimator model specifications to account for unobserved agronomic and farmer characteristics that may differ between organic and conventional fields, we cannot be sure we have isolated all sources of bias that would preclude the causal interpretation of our results. Lastly, Kern County is just one county. While the results are likely reflective of other intensive fruit and nut-producing regions in the western and southern US and similar climates, they may not reflect organic and conventional production in regions with less

valuable crop composition or lower intensity practices. Further, the overwhelming majority of organic fields in our study belonged to "mixed" farms or farms that also included conventional fields. In areas with other business structures or a greater preponderance of exclusively organic farms, perhaps there would be a different level of differentiation in pesticide use. However, for organic production to meet future food demand without large increases in farmed land will likely involve the type of intensive organic production by large "mixed" farms seen in places like Kern County.

Yields, fertilizers, water quality and quantity, soil quality, biodiversity, emissions, and price are all key aspects that are necessary for a holistic understanding of the sustainability of different production systems[11,17]. Many of these variables have been quantified and many of those, besides yields, are improved by organic practices, on a per area basis[11,17]. However, pesticide use behavior has remained obscure due to a lack of data for location and spraying patterns on organic and conventional fields. Here, we sought to elucidate the difference between organic and conventional systems with respect to the decision to spray pesticides and the decision of how much to spray. Both overall and for individual, commonly grown crops, organic fields were more likely to be "pesticide-free". However, on fields that did spray, organic and conventional fields were similar for most metrics of pesticide use, except higher acute human toxicity chemicals. Crop loss to pests is costly to the farmer and society, representing not just lost production and higher prices[59,60], but a waste of land, water, and inputs as well. More detailed toxicity and yield data, and an expanded region of study would undoubtedly provide additional insights and are worthy of future research. Nevertheless, our results suggest that while farmers seem to make the choice to spray organic fields less frequently, when they spray, they use pesticides at a similar overall rate as their conventional neighbors with substantial crop-specific heterogeneity.

## Methods

We first identify the location of organic crop fields in Kern County and then estimate whether status as organic versus conventional fields determines pesticide use (Fig. 5).

**Identifying organic fields**. We identified organic fields using a combination of California Department of Food and Agriculture (CDFA) records and Kern County Agricultural Commissioner's Office spatial data ("fields shapefiles") and pesticide use records. No single source was complete, and as such, we evaluated several different approaches to identifying organic fields.

*California Department of Food and Agriculture (CDFA) records*. Data on the location of organic fields, per the California State Organic Program, for 2013–2019 was obtained by request from the California Department of Food and Agriculture (CDFA). The CDFA, through the State Organic Program, requires annual registration of certified organic producers who have an expected gross sale of over $5000. We were specifically interested in the pesticide aspects of organic production, which is governed in our study region by the USDA's National List of Allowed and Prohibited Substances. The National List of Allowed and Prohibited Substances delineates which synthetic substances can be used and which natural substances cannot be used for pest control in US organic production. Besides substances specifically (dis)allowed on the National List, allowed substances include non-synthetic biological, botanical, and mineral inputs. Field location data were in the form of either Assessor's Parcel Number (APN) or PLS System Township-Range-Section (TRS) values, though data were reported without systematic formatting. We harmonized the CDFA APN values to merge with the Kern County Assessor's parcel shapefile (2017), which we then spatially joined with the Kern fields shapefiles. We followed a similar process with PLSS TRS values, which were then merged with the Kern County PLS Section shapefile, and joined to Kern field shapefiles. We refer to our final organic designation as "CDFA Organic". Details of the data cleaning process are described in the Ancillary Data Processing Methods section below.

*Using pesticide use reports to refine organic field identification*. After spot-checking pesticide use on CDFA Organic fields, it became clear we had not entirely

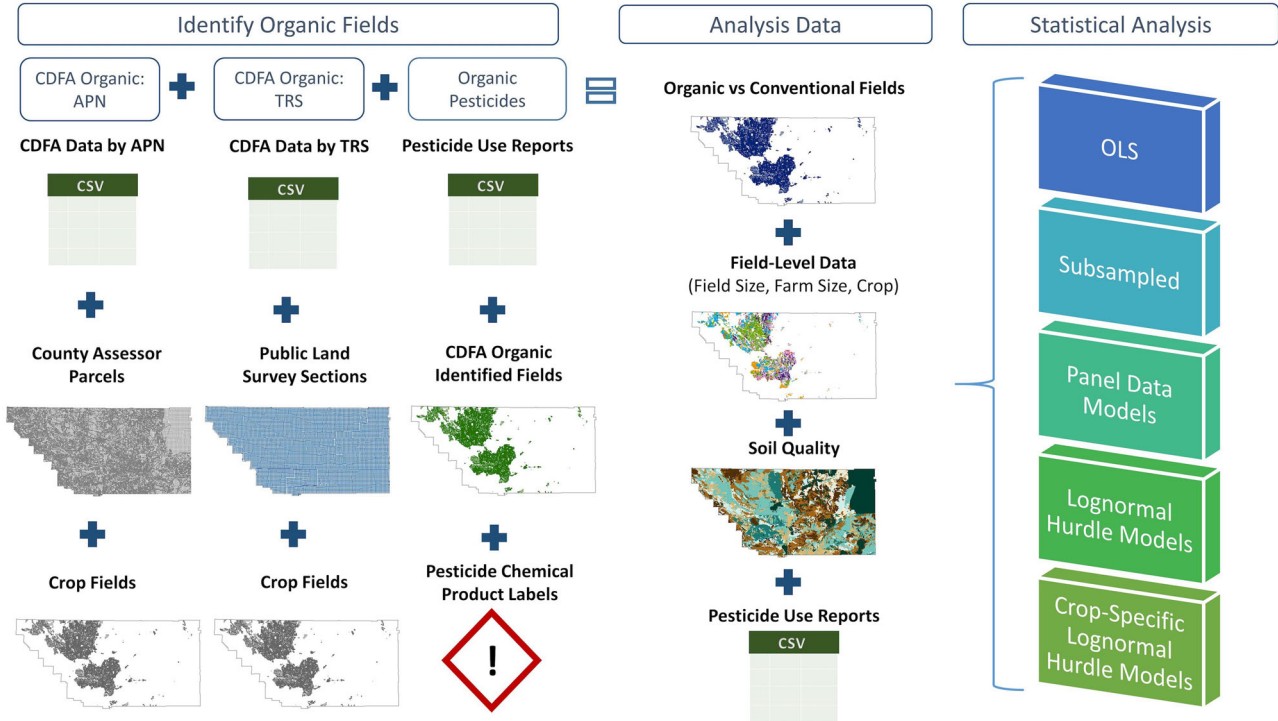

**Fig. 5 Methodology overview.** Figure outlines the main method steps from identifying organic fields to creating the analysis data to performing the statistical analyses. All images shown are simplified, visual representations of the datasets. CDFA refers to the California Department of Food and Agriculture, while APN is the Assessor's Parcel Number and TRS is the Township-Range-Section. Identifying organic fields combines the created CDFA organic APN, CDFA organic TRS, and organic pesticides data layers together to create the final organic versus conventional fields layer used in the analysis data section. All analysis data layers are then inputted into the various statistical analyses.

eliminated conventional fields. This was likely due to variation in polygon geometries between PLSS Sections, Kern County Assessor parcels, and Kern agricultural fields data. To further refine our classification, we used field-level pesticide use, again from the Kern County Agricultural Commissioner's Office. As thousands of pesticide products (active ingredients + adjuvants) are in use in Kern County, we took an iterative approach to eliminate fields using conventional pesticides. We first limited the universe of pesticides to those applied to fields that were CDFA Organic. We then identified the 50 most commonly used pesticide products by a number of applications, and manually classified each as organic or conventional. Having identified these products as described below, we matched them back in, eliminating fields that used conventional products and identifying as "PUR Organic" those that used only organic products. We repeated this process, hand identifying the most commonly used products and eliminating fields using conventional products until we had isolated fields using only organic products.

To classify a product as organic or conventional, we first searched for each product's U.S. EPA-registered product label, using the exact product name and EPA product registration number. If there was any indication on the label that the product was certified as organic by the Organic Materials Review Institute (OMRI), or said "for use in organic production" or "organic", then the pesticide was identified as organic ($n = 132$). If there was no organic indication on the product label, we searched the OMRI certification database for products with identical names and manufacturers, and identified products as organic if such certifications existed ($n = 39$). If all ingredients were defined (i.e., no inert or undefined ingredients) and were known organic active ingredients, then the pesticide was identified as organic ($n = 1$) (Supplementary Data 1). We failed to find EPA-registered labels for three products and confirmed on the California Department of Pesticide Regulation website that they are either inactive or out of production (EPA registration numbers: 52467-50008-AA-5905, 36208-50020-AA, 2935-48-AA-120). Each of the three was rarely used ($n < 4$) on CDFA organic fields across the 2013 to 2019 data. To be conservative, we defined them as conventional. If a field used no pesticides and was CDFA organic, we labeled it as organic. If it used no pesticides and was not CDFA organic, we maintained it as conventional.

*Kern County agricultural commissioner's office spatial data.* For 2017–2019, the commodity attribute field in the Kern County agricultural fields shapefiles indicated self-reported organic fields with "-ORGANIC" or "-ORG" following the commodity name. Self-reported organic crop fields may be less accurate than those identified using CDFA data as self-reporting is not validated nor required. After researching multiple large crop producers in Kern County, we suspect that many crops being grown organically with CDFA certifications are not being reported as such to the county. Still, we run a robustness test including these fields in addition to the PUR Organic fields for 2017 to 2019 to capture smaller farms that may be exempt from certifications. Doing so did not qualitatively change our results (Supplementary Figs. 2 and 3).

**Analysis data.** Beyond identifying organic fields, the goal of this project was to understand the impact of organic versus conventional production on different metrics of pesticide use. As such, we calculate various pesticide metrics, and farm, field, and crop characteristics.

*Kern County fields shapefiles.* Annual Kern County fields shapefiles are publicly available and include data on growing permit ID ("farm"), site ID, field size, crop type, date active and inactive, among other information. A crop field is defined as a permit-site-year combination. Permit numbers track individual growers through time, while site IDs are a record of each permitted site maintained by a grower each year. Site IDs occasionally persist from one year to the next (particularly for perennial crops), and they are not duplicated within the same year such that rotated crops are given unique site IDs. In some cases, multiple crops are grown on the same field simultaneously. In such cases, each crop has a unique site ID. However, when multiple crops are being grown on the same field simultaneously, the total area occupied by each crop is not specified, but rather is recorded as the total field size. To reduce bias in pesticide use rates, the field size was divided by the number of crops being grown on a field, under the assumption that all crops on multi-crop fields occupy an equal area. From crop type, we determined the crop's taxonomic family to be used in later analyses.

*Pesticide use reports.* Field-by-day-by-product level pesticide use data are available to the public on the Kern County Department of Agriculture and Measurement Standards website. Pounds of pesticide products were converted to kg of active ingredients (AI) and identified by the type of product (e.g., insecticides) and "potential hazard" to non-target organisms (e.g., bees) using the California Department of Pesticide Regulation (DPR) Product Database (see Ancillary Data Processing Methods below). We define insecticides as insecticides, insect growth regulators, miticides, and repellents, excluding products that have dual action (insecticide and fungicide), but not excluding insecticide products with adjuvant or fertilizer additives. The Product Database includes binary indicators for whether a given product is a potential hazard to different environmental outcomes. Since pesticide adjuvants ("inert" ingredients) are sometimes used in the absence of active ingredients and can nevertheless have environmental or wildlife impacts per the label, we used kg of the product rather than a kg of active ingredients as

measures of use for products of potential hazard to environment and non-target organisms. Using active ingredients instead did not qualitatively change our results (Supplementary Fig. 4). Potential hazards to non-target organisms are based on the Environmental Hazards statement on the pesticide label, which is regulated by the EPA, and based on toxicity to birds, fish, invertebrates, bees, and mammals[40]. We chose to use binary metrics rather than an aggregate index because thousands of products are used in Kern County, and toxicity metrics for the many different chemicals and numerous environmental endpoints of interest are not readily available. Thus, we proceed with measures of chemical product use ($kg\,ha^{-1}$) of potential hazard to a series of ecological and environmental endpoints (fish, aquatic species, drift, etc.). We also include products of high acute toxicity (EPA signal words 1 and 2) and lower acute toxicity (EPA signal words 3 and 4 or not required) to people. For completeness, we also aggregated ecotoxicity data for 29 products that represent about 50% of each organic and conventional pesticide use (Supplementary Table 10) from the Pesticide Products Database[61], primarily.

Using the fields shapefiles, pesticide use rates were summarized for each permit-site-year grouping for various pesticide outcomes, including $kg\,ha^{-1}$ of a pesticide product, active ingredients, and products of potential hazard to fish, bees, and/or aquatic species, products prone to drift, those categorized as insecticides, and high and low acute toxicity products, as described above. Not all environmental outcomes were equally likely, and in particular several pesticide use outcomes of potential concern lacked a sufficient number of observations on organic fields for robust comparison to their conventional counterparts (Supplementary Table 11). We statistically analyze the subset of environmental outcomes that were reasonably common in both organic and conventional fields, and thus more likely to produce reliable estimates.

*Soil spatial data.* To address the potential that organic and conventional fields have systematically different soil quality, we used the California Revised Storie Index. The Storie Index land classification system is widely used across California to assess soil quality and agricultural productivity[39] and is provided in Soil Survey Geographic Database (SSURGO) tabular data for most of the state. Ratings are systematically determined by a model in the Natural Resources Conservation Service (NRCS) National Soil Information System (NASIS) software, based on tabular data in the SSURGO database. Soil characteristics used in the model include soil profile, surface texture (e.g., loamy to clay-rich, excluding organic horizons), topographical features, growing season length, and dynamic properties (e.g., drainage, alkalinity, acidity, erosion)[39]. Fertility and other readily modified characteristics are excluded. The system uses six rating levels—" Grade One" being the highest quality soil suitable for most crops and "Grade Six" being unproductive land. We include the Storie Index as a measure of soil quality in all analyses. Further, we evaluate whether organic and conventional fields differ systematically in soil quality, using both the overall Storie Index as well as widely measured components of dynamic properties, which theoretically could be influenced by on-farm management (Supplementary Table 1).

Storie Index values were converted from a shapefile to a 60-m raster using R's fasterize package. The soil raster was used to extract Storie Index values for each Kern field polygon, using an area-weighted mean function. 319 fields had no Storie Index ratings. To assess the importance of these missing fields, we interpolated the Storie Index values using an inverse distance weighting function in R's gstat package[62,63]. The accuracy of interpolated values was checked by applying a Leave-One-Out Cross-Validation function to 500 randomly selected points. Including these fields did not qualitatively change our results.

**Statistical analysis.** Our statistical analysis proceeded in two steps. First, we evaluated whether conventional versus organic fields differed in pesticide use, modeled as a continuous variable, using pooled ordinary least squares and panel data models to determine the influence of different model specification decisions (see Ancillary Statistical Methods below, Supplementary Notes, Supplementary Tables 2 and 3). However, pesticide use can be conceived as a two-part decision. First, there is the decision to use pesticides at all, and second is the decision of how much to spray when using pesticides. Tobit models are traditionally used to estimate models with censoring. However, Tobit models force the mechanisms determining whether to spray (i.e., moving from pesticide = 0 to pesticides > 0) to be the same as the mechanisms determining the amount sprayed when some pesticides are used (pesticides when pesticides > 0). Double-hurdle models[64] are an alternative to the Tobit model that allows for the separation of these two decisions.

The mechanisms underlying the two decisions (to spray, how much to spray if spraying) can differ such that different covariates can describe each process, and the same covariates are allowed to influence the two processes in different ways (i.e., sign and magnitude can differ). The first, binary decision is usually modeled with a probit model.

$$P(y = 0|\mathbf{x}) = 1 - \Phi(\mathbf{x}\gamma) \qquad (1)$$

Then, the second decision is modeled as a linear model with pesticide use following a lognormal distribution, conditional on positive pesticide use[64]

$$\log(y)|\mathbf{x}, y > 0 \sim \text{Normal}(\mathbf{x}\beta, \sigma^2) \qquad (2)$$

where $\Phi$ is the standard normal cdf, $\mathbf{x}$ is a vector of explanatory variables including organic status, $y$ is pesticide use, and $\beta$ is a vector of coefficients. We use a

lognormal hurdle model rather than a truncated normal hurdle model since pesticide use is highly non-normal, and Q-Q plots suggested substantial model improvement using a lognormal rather than normal distribution. In contrast to the panel data models described in the Ancillary Statistical Methods below, our estimation equation used natural log-transformed variables for pesticides (and field and farm size) rather than inverse hyperbolic sine (IHS) transformation since only positive observations are included in the second hurdle model. Following insights derived from our panel data models (Supplementary Notes), we build on the basic hurdle model concept using the farm-by-crop family interaction as a random intercept in both the first and second hurdle. We chose the farm-by-crop family interaction rather than a crossed random effect due to computational feasibility with thousands of permits and hundreds of crops, due to similarity of results to the within estimator model (i.e., fixed effects in causal inference terminology; Supplementary Notes, Supplementary Table 2), and due to AIC/BIC (Supplementary Table 3). Further, we find evidence of heteroskedasticity from both visual inspection and Levine's test, which adds additional complications to computing crossed random effects. Thus, we proceed with the farm-by-crop family interaction in a random intercept model with cluster robust standard errors clustered at the same grouping. In doing so, observations, where the taxonomic family of the crop was unclear, were dropped. Of the 7367 fields that were dropped due to missing crop families, 6684 were uncultivated agriculture.

Our data are effectively repeated cross-sections rather than a true panel since fields are defined by the farm-site-year combination and thus generally change year-to-year or when crops rotate. We model it as such. This implies we do not require observations to have no spray in all time periods, as would be the case in a double hurdle panel model. Linking field IDs over time through spatial processing is complicated by crop rotations of different size areas. Since farmers may farm multiple fields under different management systems, as we illustrate here, and have different contractual obligations at a sub-farm level, requiring farms to never use pesticides on all fields is not reflective of on-the-ground decisions.

We repeated the lognormal hurdle models individually for carrots, grapes, oranges, potatoes, and onions, which were the only widely-grown crops with more than 100 organic fields. This allowed for a different slope and intercept by crop type.

We conduct several robustness checks. First, we do not have data on crop yields. However, to assess the potential implications of a yield gap on our results, we modify our per hectare rates following Ponisio et al.[15] as a robustness check. We group commodities into cereals, roots and tubers, oilseeds, legumes/pulses, fruits, and vegetables and assign them the Ponisio et al.[15] yield gap estimates for that group. Crops that did not fall into any of the above groups (e.g., cannabis) were provided the all-crop average from Ponisio et al.[15]. Second, we analyze how conventional and organic differ with respect to soil quality using a within estimator approach to account for crop-specific differences in soil quality. Third, binary toxicity metrics, while valuable given the number of chemicals and endpoints of interest here, nevertheless fail to distinguish gradations of toxicity for chemicals above (or below) the regulatory threshold. As mentioned above, the data needed to calculate many aggregate indices (e.g., Pesticide Load[57] and Environmental Impact Quotient[58]) are not readily available for all of the chemicals in our study. For completeness, we attempted to calculate the Pesticide Toxicity Index for one well-studied endpoint, fish. We supplemented data provided in Nowell et al.[41] with data from Standartox[42]. However, only about 70% of the chemicals used in our study matched, and pesticide products used on organic fields were more likely to lack toxicity information for one or more chemicals. We briefly discuss the highly preliminary investigation, given the non-random missing toxicity data.

All spatial analyses were performed in R Statistical Software v 3.5.3 and all statistical analyses were performed Stata 16 MP. For all tests, statistical significance was based on two-tailed tests with $\alpha = 0.05$.

### Ancillary data processing methods

*Cleaning parcel data.* To spatially locate organic fields, we needed to match the Assessor's parcel numbers (APNs) provided in the CDFA tabular data to APNs in the Kern County Parcel shapefile (from 2017). Over 90% of the APN entries in the CDFA data were in the format [xxx-xxx-xx], though multiple APNs were often provided in the same cell separated by line breaks, semi-colons, commas, and/or spaces. We made initial edits separating values into individual cells in Microsoft Excel since formatting was highly inconsistent. Observations whose APNs were not in the [xxx-xxx-xx] were modified so that their format matched. In the R environment, dashes were inserted after the third, sixth, and eighth characters (1234567895 became 123-456-78-95) for APNs that did not already contain them. Occasionally, APN numbers were provided with dashes, but with segments of incorrect length (e.g., 12-34-567). In these instances, APN segments were either trimmed from the right or padded with a zero on the left so they matched the [xxx-xxx-xx] format. This approach yielded the greatest number of matches and was checked for accuracy as described below. Additional segments (from APNs with more than two dashes and eight numeric characters) were dropped. A handful of APNs with fewer than eight numeric characters and no dashes were dropped entirely.

The edited CDFA APNs were then joined with the Kern County Assessor's parcel shapefile, creating the "CDFA organic shapefile". In total, 1637 of 1829 individual CDFA records joined successfully. To evaluate the accuracy of joins

between CDFA tabular data, Kern County parcel, and Kern County agricultural spatial data, we spot-checked ownership information using "Company" (CDFA) and "PERMITTEE" (Kern County agricultural data) values.

To then identify the crop fields within the organic parcels, we performed a spatial join between the CDFA organic shapefile and the Kern County fields shapefiles. Prior to performing the join, the CDFA parcels' dimensions were reduced with a 50-m buffer to eliminate spatial joins between CDFA parcels and crop fields that were only touching the parcel margins. Of five different buffer widths evaluated, 50 m reduced the number of false positives and negatives, as determined by comparing the "Company" and "PERMITTEE" values. We refer to the fields that match as "APN Organic".

*Cleaning PLSS Township-Range-Section values.* Each year several producers reported Township, Section, and Range (TRS) values, consistent with the PLS System (PLSS), rather than APN values. We used these TRS values to identify PLSS Sections that contained organic fields.

We separated any cell containing multiple TRS values and removed any prefixes such as "S", "Section", "Sec.", "T", and "R" that would prevent joining to Kern County PLSS spatial data in Excel. In the R environment, we padded the left side of the "S" value with a 0 if it was a single digit, then concatenated the three columns into a "TRS" column. We joined TRS from the CDFA tabular data to PLSS spatial data, which identified 563 Sections as containing organic fields, from 2013 to 2019, out of a total of 664 unique TRS codes in the CDFA dataset. We then performed a spatial join between PLSS Sections that contain organic fields and Kern County fields shapefiles, to identify all agriculture fields that overlap with those Sections. Additional processing using the Pesticide Use Reports is described above.

### Ancillary statistical methods

We began with a pooled ordinary least squares (OLS) model that, as the name suggests, pools observations over farms, years, and crop types. However, there may be attributes of crops or farms that may be systematically different between organic and conventional, and this systematic difference could bias our pooled OLS results. To address this, we first considered propensity score approaches but were unable to find a sufficient balance of our covariate distribution between organic and conventional fields. As an alternative, we limited our sample to fields with overlapping farmers and crop types. In other words, we focused on the subset of fields that are grown by farmers producing both organic and conventional fields and to crops that are produced both conventionally and organically. However, this shrunk our dataset by two-thirds.

To leverage more of our data, we next considered panel data models as a means to address unobserved variables. We consider both within-estimator models (also known as "fixed effects" in causal inference terminology, but different from the biostatistical use of the term) and random effects models (with random intercepts), seeking to capture characteristics of the crop, grower, and year. The advantage of a within-estimator approach is that the omitted variables are removed (through differencing) and thus, they can be correlated with covariates without biasing the estimation. In other words, pesticide use and all covariates are differenced from their crop-specific mean (or crop family, farmer, etc. specific mean, depending on the model). In doing so, the propensity for certain crops (crop family, farmer) to be grown organic or to be fast or slow adopters of new technologies is removed. The disadvantage is that characteristics shared by all fields of a crop (e.g., value) are lost in the differencing, and more importantly, that the differencing is not easily translated to nonlinear models that we employ later in the analysis. Random effects are more easily translated to nonlinear models. The disadvantage of random effects is the strong assumption that the unobserved variables are uncorrelated with the covariates[18,65], which is required for random effects coefficient estimates to be unbiased. Here, we see the difference in coefficient estimates between the within-estimator and random effects models are quite small (Supplementary Table 2).

Random effects particularly crossed random effects with thousands of permits and hundreds of crops, introduce computational challenges due to large, sparse matrices. Further, we find evidence of heteroskedasticity from both visual inspection and Levine's test, which adds additional complications to computing crossed random effects. We proceed using the farm-by-crop family interaction in a random intercept model with cluster robust standard errors clustered at the same grouping based on AIC/BIC (Supplementary Table 3), computational feasibility, and similarity to the within-estimator results (Supplementary Table 2). Observations, where the taxonomic family of the crop was unclear, were dropped in any models including family in either the random effects or the cluster robust standard errors. Of the 7367 fields that were dropped due to missing crop families, 6684 were uncultivated agriculture.

In the panel data models, we used IHS transformations to accommodate highly non-normal pesticide (and field and farm size) data. IHS is very similar to natural log transformation[66] but is defined at zero, which is important given a sizable fraction of our observations have zero pesticide use. As with log–log transformations, IHS–IHS transformation can be interpreted as elasticities. We pre-multiply pesticide use by 100 to improve estimation[66], though this does not affect interpretation. As described above, we leverage insights on model specification from the panel data models, but rely on the double hurdle models to parse apart the decision to spray from the decision of how much to spray.

**Reporting summary**. Further information on research design is available in the Nature Research Reporting Summary linked to this article.

## Data availability

The agriculture and pesticide use data for Kern County are available at http://www.kernag.com/gis/gis-data.asp and Kern County parcel data is available at https://geodat-kernco.opendata.arcgis.com/. The California Department of Pesticide Regulation Product Database is available at https://apps.cdpr.ca.gov/docs/label/labelque.cfm. Soil quality data are from Natural Resource Conservation Service SSUGO data available from https://websoilsurvey.nrcs.usda.gov/ and accessed using https://websoilsurvey.sc.egov.usda.gov/App/WebSoilSurvey.aspx. Data on registered organic producers was obtained through request from the California State Organics Program, https://www.cdfa.ca.gov/is/organicprogram/. Hand classified organic and conventional pesticides are provided in Supplementary Data File 1. Data to repeat the main analyses is available on Dryad, https://doi.org/10.25349/D9Q02T. Regression tables underlying all analysis figures (main text, supplementary) are provided in the Supplementary Information.

## Code availability

Code to repeat the main analyses is available in Supplementary Data 2.

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

## Acknowledgements
We acknowledge the California Department of Pesticide Regulation, California Department of Food and Agriculture, and the Kern County Department of Agriculture and Measurement Standards for curating and sharing data without which this project would not be possible. We thank S. Philpott, N. Parker, and A. MacDonald for comments on an earlier draft, and M. Patton for research assistance. We acknowledge the helpful comments of the editor and reviewers, and the editor for the recommendation to move all Supplementary Methods to the main text. AEL acknowledges NSF DEB 2042526, LCP acknowledges the University of Colorado Boulder's Interdisciplinary Quantitative Biology certification program, S.M. acknowledges the UBC Four Year Doctoral Fellowship (4YF).

## Author contributions
A.E.L. conceived the study, conducted the statistical analyses, and drafted the paper, L.C.P. and S.M. processed the data and drafted the paper. All authors contributed substantially to revisions.

## Competing interests
The authors declare no competing interests.
