## [Peer Review File · Nature Communications]

Reviewers' Comments:

Reviewer #1:

Remarks to the Author:

This paper is an interesting and appropriate synthesis which questions a piece of 'popular perception' and I'm excited to see it. The organic-conventional debate comes up periodically in the literature because of the ongoing greenwashed marketing of this industry- with scientists periodically popping up and pointing out that agricultural systems are a bit more complicated than this discrete comparison, that attributes associated with one type of agriculture are not binary features, etc. This study tackles the 'pesticide free' association with organic cropping systems and finds that, while organic farms are less likely to use pesticides in general, those who do use just as much as conventional farmers. The technical approach, between the data collection and the statistical models used, are appropriate, and represent a huge undertaking of data collation. The statistical approach leans into econometrics but the mechanics are very well described and mathematically plausible (even to an ecologist!)

I'm having a bit of trouble with the introduction because it's not as clearly separating the difference between pesticide use (yes/no), pesticide rate (how much?), frequency (how often?) and pesticide toxicity (What's applied and who it affects?). The modelling approach mainly separates the first question from the latter. One of the main questions that arises is, if organic farmers who use pesticides are using the same amount of pesticides (or same number of applications?) as conventional farmers, then, relative toxicity of the products used will be a major factor in determining total impacts. For example, although it's certainly not toxicologically neutral (or why would a farmer use it for pest control?) a dormant oil, gram for gram, won't have the same relative toxicity as an organophosphate, but these products are also typically used at dramatically different rates. This idea itself may be outside the scope of the present study but I think it's an important concept to address, at least through literature examples in the paper. This becomes very important to address in the introduction in a paper of this nature because I expect many readers will not dive deeply into the methods.

I also have a few line comments:

L14: Is it necessary to specify the statistical tool used in the abstract? I find if it's not a core, unique aspect of the story, it just weighs down the piece with technobabble. I see the particular modeling approach as a tool to get *at* the story.

L163- This is the material I was hoping to see- can the materials of this section be developed a bit more? That sentence stating on 163 is doing a lot of work without a lot of specific detail about how this is done or what it means

L188- Verb tenses seem to be changing?

L276- a bit repetitive

L278- Now this is interesting- are there differences in pesticide use on organic fields based on their context? I.e. if a farmer grows mixed organic/conventional crops, are they more likely to use organic pesticides than a farmer that only grows organic crops- is it an ideological or business decision?

Reviewer #2:

Remarks to the Author:

With a view to identify the environmental benefits and drawbacks of different production systems, the paper presents a comparative analysis between conventional and organic agriculture to establish the likelihood of the decisions in organic agriculture about spraying pesticides or not, and if so, how much to spray?

Based on the secondary data and information on some 9000 organic fields during the period of 2103-2019 (from a single but important county in California, USA), the authors conclude (through modelling using lognormal hurdle models) that:

- (i) the fields under organic agriculture are less likely to be sprayed with pesticides, and
- (ii) the organic fields that are sprayed are likely to receive similar levels of pesticides as the farms under conventional agriculture, in the neighbourhood.

In reaching these conclusions, the authors have given considerations to a range of factors, such as

the spatial location of organic crop fields, field-level crop and pesticide use data, field size, farm size and soil quality together with agronomic practices of the farmers. In this process, the authors have also provided significant insights into several additional aspects of organic farming, such as their sizes, location (e.g. on better soils), crop types and pesticide use pattern.

The subject matter presented in the paper is of global interest. The findings are of significant interest not only to the scientists seeking answers to questions related to pesticides and sustainability of agricultural production but also to the natural resource managers & policy makers engaged in environmental sustainability. Besides the general community at large is likely to be interested in the topic and especially in the implications of the findings from this study.

General comments:

Generally, the work appears of high quality and to be convincing. The authors have adopted rigour in their methodology and to make the most out of the available data, despite some limitations and also the lack of data. I must confess that my expertise in the modelling methods and approach used here is limited and I am unable to do a thorough analysis of the finer methodological details as well as the assumptions made in the study. Nevertheless, the authors seem to have tested the validity of their assumptions during their analyses and accordingly have taken the best course possible. It seems like a sound study, albeit focussed on a single county. Therefore, the authors need to be congratulated on this work.

However, I have several questions and some concerns about the approach the authors have taken, especially in evaluation of toxicological impact of pesticides. Assuming the study methodology is sound, my questions are of higher level.

Let us first consider the conclusions from this study. The key conclusions include: "...on average, organic fields are much more likely to be "pesticide free", and thus lack ecotoxicological impacts or residues stemming from their own pesticide use" (see lines 298-299, also line 17 in the abstract). The first conclusion is based on the 18-31 percentage point reduction in the probability of spray under organic farming that they found in their analyses. Statistically how robust are these numbers, what is the confidence interval around these numbers? Based on the level of significance of their results, could the authors justifiably make such a claim and to this degree (...much more likely to be "pesticide free"...). This is a crucial point. Therefore, the authors need to make a more convincing argument and provide further justification to this statement (line 298-299). I can imagine a headline quotation from this work that could easily be taken out of context (of a single country in USA) in popular media and used more broadly for vested interests.

The second phrase in the statement (lines 298-299) i.e. "...and thus lack ecotoxicological impacts..." is also concerning. This is based on their observation (line 301) "...we see a significant 27 % reduction in pesticide products (kg ha⁻¹) of high acute toxicity for organic system versus conventional. For products of potential hazard to fish, etc., generally did not see a significant reduction." Given the binary nature of data used, is this conclusion sound? The margin seems too small to claim this.

Furthermore, I am not sure if the toxicological impact has been evaluated with the best available approach. As stated in the methodological section (line 471-472), the authors have used binary indicators i.e. whether the product is a potential hazard or not. In Table S10, I see the toxicological parameters for fish, bees and invertebrates have been given but alga is not included. Yet the list of products being used under both systems include many products that are herbicides which are expected to be toxic to algae and may or may not be toxic to vertebrates or bees. The omission of this taxa weakens the study.

Usually in assessment of ecosystem impact of pesticides different trophic levels are considered, such as algae (representing the base of food chain); water flea (representing invertebrates) fish (vertebrates) and Bees. As far as I know the product label does not have enough information to indicate the toxicity to different organisms. However, USEPA has a database of toxicity parameters in separate databases (<https://cfpub.epa.gov/ecotox/>) that can be useful in such assessment. In recent years, a considerable advancement appears to have been made by life-cycle impact assessors on how to characterise toxicological impact of pesticides. For example, the work by UNEP and SETAC Life Cycle Scientists leading to a scientific consensus model (USEtox) that reduced the intermodal variation by several orders of magnitude (Rosenbaum et al. 2008 - DOI 10.1007/s11367-008-0038-4). The model allows the chemical impact characterisation related to human and freshwater ecotoxicity. This approach has been recently used successfully by others for both human and ecotoxicological hazard assessment (e.g. Navarro et al. 2021 - Environ. Sci. Technol. 2021, 55, 1290–1300 <https://dx.doi.org/10.1021/acs.est.0c05717>). Therefore, as a minimum, the authors need to justify that they have used the best available approach for

toxicological assessment.

Conventional agriculture is also not a homogenous farming method – there are many variants in terms of agronomic practices, some of which are directly driven by environmental considerations – such as Integrated Pest Management. Has that been considered and if not, how this is likely to impact the comparative analysis and especially the conclusion that there is little difference between sprayed farms under organic versus conventional agriculture?

Another area of concern is the largescale applicability of the findings. Given that a single county data has been used, first question is how valid these findings are likely to be for other regions in USA and more importantly globally? As mentioned earlier, due to strong interest in this topic there is considerable scope for “out of context’ utilisation of conclusions drawn here. In the discussion sections authors highlight (lines 339 onwards) several limitations and caveats. However, they do not clearly state the broader applicability of these findings nationally and internationally. I had expected a much greater difference between organic and conventional practices, and so would the community, I think. Can the authors also comment on the extent of reduction observed here, in the context of community expectations of organic farming?

Finally, given the outreach of the Nature Communication the manuscripts are expected to be newsworthy. The current manuscript suffers from (a) very heavy subject matter (mathematically oriented) is likely to be understood by only a small group of readers of the journal and (b) the authors need to make the methodology also as easy as possible for the readers.

Specific comments

Lines 278-282

Since, organic fields, while smaller in size, tend to be part of farms far larger than their conventional counterpart. Does that not influence the decision about what to spray and how much to spray compared to a farm which is exclusively organic. The exclusive organic farmers may be at a greater liberty to choose products with careful considerations in terms of both their environmental footprint and suitability for organic farming. I realise that there are certain products that are certified for organic farming.

Line 249-252-

“A small number of observations (n = 319) were dropped due to missing soil quality data. Including observations with interpolated soil quality has little effect on our results (Figure S2).” Given this observation (soil quality not being a sensitive parameter) should these have been included in the analyses and would that have been beneficial in the second hurdle, such as in the case of self-reported organic (line 252)? Was this tested?

Line 313-322...

In the absence of disaggregated yield data, availability of which “... would dramatically improve understanding of the environmental impacts of organic production” (line 321-322), how seriously could it undermine the current conclusions.

Rai Kookana

Reviewer #3:

Remarks to the Author:

The study represents a systematic comparison of pesticide use in organic and conventional horticulture in one county in California. It is based on sound econometric analysis and detailed field-level data, covering a large number of observations. The results are discussed in the light of the current literature and methodological strengths and weaknesses are explained. The topic has a high policy relevance and can inform the debate around sustainable intensification.

General remarks/questions:

It seems ok to first pool all observations and then run crop specific regressions. I would however make sure that the crop-specific estimation and the pesticide risk metrics are the main source for conclusions and policy recommendation. The study make a considerable effort to address selection bias. The readers need to be made aware that it can also introduce aggregation bias if grapes and potatoes are pooled. To get an idea of the data behind each crop-metric specific estimation it would be very useful to have the number of observations indicated in the figures (which otherwise are very clear and informative). Also averaging the crop-specific results and comparing that to the

pooled estimation results would be helpful.

An unambiguous definition of pesticide use in organic farming as opposed to pesticide use in conventional farming is needed. What type of plant protection products are allowed for organic farming in the study site?

Why is there a need to make the study spatially explicit? It is not entirely clear why the pesticide use reports are not sufficient for performing the statistical analysis. Is this because there is no information about the certification status of a field in the pesticide use reports? It should be better explained how relevant the results are beyond the study region.

Specific remarks/questions:

Rows 9-10: I agree that in terms of pesticide use there is a knowledge gap. However, I would not say that many of the environmental impacts of organic farming are poorly understood. There is a rich literature on comparative sustainability assessments. Per unit area, the evidence for many environmental impacts is quite clear. Per unit output there tends to be much more uncertainty. Figure 1 in this paper by Ramankutty et al. gives a good impression of the available evidence: <https://onlinelibrary.wiley.com/doi/abs/10.1111/agec.12534>.

Rows 22-23: "...but organic fields that are sprayed tend to receive similar levels of pesticides as their conventional neighbors." This can be misunderstood. The overall levels might be the same, but what does this information really mean? Active ingredients differ and pesticide risks should be the most policy-relevant metric here.

Rows 31-34: A publication with more recent data (until 2019) is available. Data from the 2021 World of Organic Agriculture report puts the area under organic farming at 1.5% and 73.3 million ha. See the report here: <https://www.fibl.org/fileadmin/documents/shop/1150-organic-world-2021.pdf>

Row 37: I suggest to mention the EU's Farm to Fork strategy here in addition to the paper from 2009. This is a recent paper on organic farming in the context of the F2F strategy: <https://onlinelibrary.wiley.com/doi/abs/10.1111/1746-692X.12294>

Rows 42-52: This is definitely true when studying actual behaviour. I would clarify that this is the intention here. It would still be important to also mention that there are long-term trials that compare organic and conventional cropping and that control for agro-ecological baseline conditions and management differences.

Row 65: Which regulation are you referring to? There are multiple organic standards, which have different requirements. It would be good to check the COROS (Common Objectives and Requirements for Organic Standards) by IFOAM. In my understanding no chemical substances are allowed in organic farming if it is an officially recognised standard. Copper is a mineral substance not a chemical pesticide. Of course it is still hazardous.

Row 67: I suggest to give more examples of a hazardous active ingredient in organic farming. Many know that copper is harmful, but it would be good for readers to be aware also of other active ingredients that are used in organic farming and that are (potentially) harmful to humans and the environment.

Rows 89-94: Some more information on the sampling strategy would be helpful. From what population was the sample drawn? How were sample fields or farms selected? Or are all farms and fields in the county included?

Rows 123-124: How did you control for these variables?

Rows 128-131: It is not fully clear that subsetting the data allows fully controlling for farm and farmer characteristics? Are there for example farms that operate different enterprises for organic and conventional vegetable production or that have different managers depending on the production system? Also, by reducing your sample to farms with both types of fields your results

are no longer representative for the larger population of horticultural farms in the region. It would be interesting to perform the analysis also using the entire dataset and relying on reweighting methods to control for selection bias. This should of course not substitute the present analysis, but it can be an interesting robustness check. You mention that the use of propensity score matching does not lead to sufficient balance. Entropy weights are an appealing alternative, because of the exact adjustment of covariate moments. For more information see <https://web.stanford.edu/~jhain/Paper/PA2012.pdf>.

Row 132: You wrote earlier that the number of organic fields ranged from a low of 936 in 2013 to a high of 1,544 in 2017. How did you account for conversion to organic or abandoning certification in your panel data model?

Rows 152-155: A hurdle model seems a plausible choice here. It would be good to explain that this model choice implies that zeros represent the true choice of the farmers rather than a decision that the farmer had no control over for some reason.

Rows 161-170: It is definitely true that overall pesticide use quantity is not a good indicator of how hazardous an active ingredient is for the environment and humans. It needs to be justified why the EPA pesticide label information is good enough to quantify harm. There are a large number of metrics available that measure this, e.g. the Environmental Impact Quotient, the Pesticide Load Index or the Environmental Yardstick. The following paper by Möhring et al. illustrates the relevance of pesticide indicator choice for the results: <https://onlinelibrary.wiley.com/doi/full/10.1111/agec.12563> . If it is difficult to calculate a more refined index because of the diversity of active ingredients used, it would be interesting to know how robust the EPA information is against these more advanced metrics for the 10 most important active ingredients in the region.

Figure 2: Why is the change in terms of high toxicity product use not reflected in the bee and fish hazard indicators? Does high toxicity only refer to humans in this case?

Rows 227-228: This is an interesting finding. What drives this massive increase in use?

Row 276: What does intrinsically better soil refer to? Soil type?

Rows 308-309: This shows that recommendations primarily need to be crop-specific. In addition, pesticide risk rather than use levels for individual crops would be the most valuable information for designing capacity development interventions and policies.

Rows 361: Could you provide some examples where you expect similar production conditions and production systems than in Kern County. This would be helpful to get a better idea in how far the results also apply to other regions.

We sincerely appreciate the insightful comments provided by the three reviewers. Our manuscript is clearer and more robust as a result of the review process. Thank you. Please find the original comments in italics, followed by the responses indented below.

REVIEWER COMMENTS

Reviewer #1 (Remarks to the Author):

This paper is an interesting and appropriate synthesis which questions a piece of ‘popular perception’ and I’m excited to see it. The organic-conventional debate comes up periodically in the literature because of the ongoing greenwashed marketing of this industry- with scientists periodically popping up and pointing out that agricultural systems are a bit more complicated than this discrete comparison, that attributes associated with one type of agriculture are not binary features, etc. This study tackles the ‘pesticide free’ association with organic cropping systems and finds that, while organic farms are less likely to use pesticides in general, those who do use just as much as conventional farmers. The technical approach, between the data collection and the statistical models used, are appropriate, and represent a huge undertaking of data collation. The statistical approach leans into econometrics but the mechanics are very well described and mathematically plausible (even to an ecologist!)

Thank you.

I’m having a bit of trouble with the introduction because it’s not as clearly separating the difference between pesticide use (yes/no), pesticide rate (how much?), frequency (how often?) and pesticide toxicity (What’s applied and who it affects?). The modelling approach mainly separates the first question from the latter. One of the main questions that arises is, if organic farmers who use pesticides are using the same amount of pesticides (or same number of applications?) as conventional farmers, then, relative toxicity of the products used will be a major factor in determining total impacts. For example, although it’s certainly not toxicologically neutral (or why would a farmer use it for pest control?) a dormant oil, gram for gram, won’t have the same relative toxicity as an organophosphate, but these products are also typically used at dramatically different rates. This idea itself may be outside the scope of the present study but I think it’s an important concept to address, at least through literature examples in the paper. This becomes very important to address in the introduction in a paper of this nature because I expect many readers will not dive deeply into the methods.

Thank you for the comment. In direct response to this comment, we have added several lines and citations in the introduction clarifying use and toxicity, and acknowledging that they are not synonymous (lines 51-60, 76-83).

Additionally, in response to this comment and that of the other reviews, we re-evaluated continuous measures of toxicity. The data needed to calculate many toxicity metrics (e.g. Pesticide Load, Pesticide Toxicity Index) are not always available. We tried to focus on an endpoint with substantial data, and at least compare the binary metrics of

toxicity to a continuous metric for one endpoint. We chose fish, based on Nowell et al. 2014, who provided exceptional appendices for many chemicals in use in the US. We supplemented the Nowell et al. (2014) with data from Ecotox using the R package standartox (Scharmüller 2021). While we tried to follow the Nowell et al. (2014) methods, our mean toxicity concentrations differed for chemicals that existed in both the Nowell et al. (2014) data and our data. This was likely due to the treatment of outliers and various filters of studies included in the Nowell et al. (2014). As we are not toxicologists, we deferred to the Nowell et al. (2014) data where available, and used our estimates built with standartox where Nowell et al. (2014) were lacking data. Still, only about 70% of chemicals in use in our study period matched. Additionally, many products, which can include multiple chemicals, did not have full toxicity information. As the Pesticide Toxicity Index, as with other indices, is a weighted sum, those products without toxicity were not counted at all and products with chemicals missing toxicity were underestimated. Products used on organic fields were more likely to lack toxicity information for one or all chemicals, which biases our analysis. While chemicals lacking toxicity may be lower toxicity chemicals, we do not have the evidence or the expertise to support that claim.

Noting the bias due to non-random missing toxicity data, we nevertheless tried to run our double hurdle model for fields that had products with toxicity information for, on average, 70% of the chemicals. Just 35% of our organic fields that had pesticide use met this condition, while 67% of the conventional fields with pesticide use met this condition. Our preliminary investigation suggests a decrease in toxicity weighted use (for fish) of about 50% for fields that are organic relative to conventional, depending on whether we adjust for yield gaps or not. Given the data limitations, we feel this is suggestive and worthy of future investigation, and we note the approach, highly preliminary results and limitations in the methods (lines 615-624), results (lines 254-271) and discussion (lines 375-378).

In revisiting our analysis to address the toxicity comment here and below, we noticed an updated version of the product table for 2018. As 2019 isn't yet available, we match 2019 pesticide use to the most recently available table. This update resulted in a small number of rarely used chemicals (21/1100+) merging that didn't before, leaving just 1 chemical used one time in our 2019 pesticide use data, but not in the 2018 product database. We reran the analysis from start to finish, which had extremely small impacts on our coefficient estimates (usually in the hundredths or thousandths place). We've updated all data figures and tables.

I also have a few line comments:

*L14: Is it necessary to specify the statistical tool used in the abstract? I find if it's not a core, unique aspect of the story, it just weighs down the piece with technobabble. I see the particular modeling approach as a tool to get *at* the story.*

We agree. We have removed mention of the statistical tool and have shortened the abstract to the required 150 words.

L163- This is the material I was hoping to see- can the materials of this section be developed a bit more? That sentence stating on 163 is doing a lot of work without a lot of specific detail about how this is done or what it means

We have expanded the discussion of the binary thresholds in the introduction (lines 76-82), methods (lines 512-519) and results (lines 143-150). The results section, which is the focus on this comment, now reads:

“The pesticide label, which is governed by the EPA, includes hazards statements that reflect whether one or more chemicals in the product exceeds thresholds regarding acute toxicity to humans, as well as for different environmental and ecological outcomes. These statements are generally based on acute toxicity studies for humans, and for birds, fish, invertebrates, bees, and mammals (see methods). Additionally, labels include a statement based on the product’s potential to drift or be transported in other media, and information on the active ingredients from which target taxa can be derived, among other information.”

L188- Verb tenses seem to be changing?

Thank you. We changed it to the simple present tense to match the rest of the results.

L276- a bit repetitive

We reworded the line in question to reduce repetition.

L278- Now this is interesting- are there differences in pesticide use on organic fields based on their context? I.e. if a farmer grows mixed organic/conventional crops, are they more likely to use organic pesticides than a farmer that only grows organic crops- is it an ideological or business decision?

We revisited our data with this interesting comment in mind. In our study region, there are very few organic fields (<3%) that belong to exclusively organic farms. Rather, the vast majority of organic fields are part of farms that include both organic and conventional fields. Based on basic summary statistics, field size, soil quality and average pesticide use per hectare is similar among those grown by exclusively organic versus mixed farms. However, we do not observe organizational structure, so it is possible that those that appear exclusively organic are a subsidiary of a larger producer that also produces conventional agriculture.

Reviewer #2 (Remarks to the Author):

With a view to identify the environmental benefits and drawbacks of different production systems, the paper presents a comparative analysis between conventional and organic

agriculture to establish the likelihood of the decisions in organic agriculture about spraying pesticides or not, and if so, how much to spray?

Based on the secondary data and information on some 9000 organic fields during the period of 2103-2019 (from a single but important county in California, USA), the authors conclude (through modelling using lognormal hurdle models) that:

*(i) the fields under organic agriculture are less likely to be sprayed with pesticides, and
(ii) the organic fields that are sprayed are likely to receive similar levels of pesticides as the farms under conventional agriculture, in the neighbourhood.*

In reaching these conclusions, the authors have given considerations to a range of factors, such as the spatial location of organic crop fields, field-level crop and pesticide use data, field size, farm size and soil quality together with agronomic practices of the farmers. In this process, the authors have also provided significant insights into several additional aspects of organic farming, such as their sizes, location (e.g. on better soils), crop types and pesticide use pattern. The subject matter presented in the paper is of global interest. The findings are of significant interest not only to the scientists seeking answers to questions related to pesticides and sustainability of agricultural production but also to the natural resource managers & policy makers engaged in environmental sustainability. Besides the general community at large is likely to be interested in the topic and especially in the implications of the findings from this study.

Thank you.

General comments:

Generally, the work appears of high quality and to be convincing. The authors have adopted rigour in their methodology and to make the most out of the available data, despite some limitations and also the lack of data. I must confess that my expertise in the modelling methods and approach used here is limited and I am unable to do a thorough analysis of the finer methodological details as well as the assumptions made in the study. Nevertheless, the authors seem to have tested the validity of their assumptions during their analyses and accordingly have taken the best course possible. It seems like a sound study, albeit focussed on a single county. Therefore, the authors need to be congratulated on this work.

Very much appreciated.

However, I have several questions and some concerns about the approach the authors have taken, especially in evaluation of toxicological impact of pesticides. Assuming the study methodology is sound, my questions are of higher level.

Let us first consider the conclusions from this study. The key conclusions include: "...on average, organic fields are much more likely to be "pesticide free", and thus lack ecotoxicological impacts or residues stemming from their own pesticide use" (see lines 298-299, also line 17 in the abstract).

The first conclusion is based on the 18-31 percentage point reduction in the probability of spray under organic farming that they found in their analyses. Statistically how robust are these numbers, what is the confidence interval around these numbers? Based on the level of

significance of their results, could the authors justifiably make such a claim and to this degree (...much more likely to be “pesticide free”...). This is a crucial point. Therefore, the authors need to make a more convincing argument and provide further justification to this statement (line 298-299). I can imagine a headline quotation from this work that could easily be taken out of context (of a single country in USA) in popular media and used more broadly for vested interests.

Thank you for noting this concern. We agree. While the results are robust to different model specifications and different outcomes, there is a confidence interval around each number. As the reviewer hints at, the confidence interval around the point estimates excludes zero, but is not reflected in the presentation of point estimates alone. We do include the standard error as a measure of the spread in the results section. However, to avoid confusion in the discussion or the results being taken out of context, we have reworded these sentences, lines 311-313. We hope this provides a more nuanced description.

The second phrase in the statement (lines 298-299) i.e. “...and thus lack ecotoxicological impacts...” is also concerning. This is based on their observation (line 301) “...we see a significant 27 % reduction in pesticide products (kg ha⁻¹) of high acute toxicity for organic system versus conventional. For products of potential hazard to fish, etc., generally did not see a significant reduction.” Given the binary nature of data used, is this conclusion sound? The margin seems too small to claim this.

We apologize for our messy language. We were referring to the greater propensity of organic fields to be pesticide-free and thus lack ecotoxicological impacts on those fields that were not sprayed. We have reworded the sentence, lines 311-313.

Furthermore, I am not sure if the toxicological impact has been evaluated with the best available approach. As stated in the methodological section (line 471-472), the authors have used binary indicators i.e. whether the product is a potential hazard or not.

Thank you for the comment. To address toxicity comments of this reviewer and the others, we sought to develop a continuous metric of toxicity. We considered the approaches the reviewer suggested in the paragraph below, but data were not readily available for all of our chemicals. We tried to develop the Pesticide Toxicity Index (Nowell et al. 2014) for fish, as it is one of the most studied taxa, at least for the chemicals in use in Kern County. As described in response to Reviewer 1, we supplemented the Nowell et al. (2014) with data from Ecotox using the R package standartox (Scharmüller 2021). Only about 70% of chemicals in use in our study period matched. Additionally, many products, which can include multiple chemicals, did not have full toxicity information. As the Pesticide Toxicity Index, as with other indices, is a weighted sum, those products without toxicity were not counted at all and products with chemicals missing toxicity were underestimated. Products used on organic fields were more likely to lack toxicity information for one or all chemicals, which biases our analysis.

Nevertheless, we tried to run a double hurdle model for the subset of fields that had products with toxicity information for, on average, 70% of the chemicals. Just 35% of our organic fields that had pesticide use met this condition, while 67% of the conventional fields with pesticide use met this condition. Our preliminary results suggest a decrease in toxicity weighted use (for fish) of about 50% for fields that are organic relative to conventional, depending on whether we adjust for yield gaps or not. Given the data limitations, we feel this is suggestive and worthy of future investigation, and we note the approach, highly preliminary results and limitations in the methods (lines 615-624), results (lines 254-271) and discussion (lines 375-378).

In Table S10, I see the toxicological parameters for fish, bees and invertebrates have been given but algae is not included. Yet the list of products being used under both systems include many products that are herbicides which are expected to be toxic to algae and may or may not be toxic to vertebrates or bees. The omission of this taxa weakens the study. Usually in assessment of ecosystem impact of pesticides different trophic levels are considered, such as algae (representing the base of food chain); water flea (representing invertebrates) fish (vertebrates) and Bees.

Thank you for the comment. We have added toxicity to algae in Table S10.

As far as I know the product label does not have enough information to indicate the toxicity to different organisms. However, USEPA has a database of toxicity parameters in separate databases (<https://cfpub.epa.gov/ecotox/>) that can be useful in such assessment.

We elaborate on the information available on the pesticide label. In particular, we discuss the information in the Environmental Hazards statement and what toxicity tests those are based on in the introduction (lines 76-78), methods (lines 512-519) and results (lines 143-150). Nevertheless, we agree that binary metrics of toxicity are limiting and ECOTOX and other toxicity databases can be useful. As noted above, we used standartox R package to access ECOTOX data, and tried to repeat the median toxic concentration described in Nowell et al. (2014). However, we were not able to repeat their analysis exactly, likely due to the treatment of outliers and/or other details that were applied to aggregate the many studies available in ECOTOX. For one crop or a few dozens of chemicals, a far more complete ecotoxicological assessment is possible. However, in our study location and time period there are thousands of products and hundreds of chemicals in use. Because we are trying to understand how organic vs conventional impacts (toxicity-weighted) pesticide use, missing toxicity data is a problem. It is particularly a problem because it does not appear to be missing randomly, and thus could bias our analysis.

In recent years, a considerable advancement appears to have been made by life-cycle impact assessors on how to characterise toxicological impact of pesticides. For example, the work by UNEP and SETAC Life Cycle Scientists leading to a scientific consensus model (USEtox) that reduced the intermodal variation by several orders of magnitude (Rosenbaum et al. 2008 - DOI 10.1007/s11367-008-0038-4). The model allows the chemical impact characterisation related to

human and freshwater ecotoxicity. This approach has been recently used successfully by others for both human and ecotoxicological hazard assessment (e.g. Navarro et al. 2021 - Environ. Sci. Technol. 2021, 55, 1290–1300 <https://dx.doi.org/10.1021/acs.est.0c05717>). Therefore, as a minimum, the authors need to justify that they have used the best available approach for toxicological assessment.

Thank you for the comment. We described above why we landed on binary metrics and elaborate throughout the paper on the utility and limitations of our approach. For the large number of chemicals and endpoints in our study, a comprehensive analysis of toxicity was unfortunately out of scope. As noted above, we tried to evaluate toxicity to fish following Nowell et al., (2014), though were not entirely successful. Like Navarro et al. 2021, gaps remain in the toxicity data that Nowell et al. (2014) used and which we tried to extend. Given Nowell et al. (2014) sought to determine toxicity for all chemicals in use in US agriculture, and we supplemented that with data from standartox, which was published in 2021, it is unlikely other models would provide more comprehensive coverage of our study area. As we note in the discussion, it would be an extremely valuable advance to include a continuous metric of toxicity for different endpoints to further differentiate the environmental impacts of pesticide use decisions on organic and conventional fields.

Conventional agriculture is also not a homogenous farming method – there are many variants in terms of agronomic practices, some of which are directly driven by environmental considerations – such as Integrated Pest Management. Has that been considered and if not, how this is likely to impact the comparative analysis and especially the conclusion that there is little difference between sprayed farms under organic versus conventional agriculture?

We agree conventional agriculture is not homogeneous. Farmers may use a number of IPM techniques, and adoption of IPM is widespread in fruit, nut and vineyards in California (Farrar et al. 2016). Unfortunately, we do not observe specific IPM practices in our data. However, we do include farmer-by-crop family random effects (random intercepts) to account for the potential idiosyncratic differences in IPM adoption or other agronomic practices that an individual farmer may adopt on specific types of crops. This would not be sufficient if, for example, a given farmer engaged in more pest monitoring (e.g. scouting) or pest suppression (e.g. ground covers or biological pest control) on their organic fields than their conventional fields. In that case, presuming IPM led to less pesticide use (or use with specific environmental hazards) than would otherwise occur, we may be overestimating the difference in pesticide use between organic and conventional fields.

Another area of concern is the largescale applicability of the findings. Given that a single county data has been used, first question is how valid these findings are likely to be for other regions in USA and more importantly globally? As mentioned earlier, due to strong interest in this topic there is considerable scope for “out of context” utilisation of conclusions drawn here. In the discussion sections authors highlight (lines 339 onwards) several limitations and caveats.

However, they do not clearly state the broader applicability of these findings nationally and internationally.

Thank you for the comment. We have added discussion (lines 382-391) to describe the applicability of our findings. We expect our findings reflect other intensive, high-value regions, but are unlikely to apply to regions with lower value crops (e.g. cereals, soybeans) that have far reduced levels of inputs or regions with lower intensity practices. However, we humbly suggest intensification of organic practices is a likely outcome if organics are to start producing a greater share of global food production.

I had expected a much greater difference between organic and conventional practices, and so would the community, I think. Can the authors also comment on the extent of reduction observed here, in the context of community expectations of organic farming?

We agree. Admittedly we did as well. We anticipated a difference in the propensity to spray based on prior experience with the pesticide use data. However, we did not expect the difference in use to be so similar on fields that did use pesticides, particularly for use by toxicity category. We believe this reflects the intensity of production in this region. Very few fields are part of farms that are entirely organic, rather the majority of organic fields are part of farms that produce both organic and conventional crops. Thus, we believe it is very much a market driven organic production landscape, rather than the type of farming many consumers may associate with the organic label.

Finally, given the outreach of the Nature Communication the manuscripts are expected to be newsworthy. The current manuscript suffers from (a) very heavy subject matter (mathematically oriented) is likely to be understood by only a small group of readers of the journal and (b) the authors need to make the methodology also as easy as possible for the readers.

We agree. As noted by both this reviewer and Reviewer 1, the methods are a tool and not necessarily the novelty. We have removed mention of the methods from the abstract. We have also shortened the methods to focus on our main models so the reader is not bombarded with several approaches that are ancillary to our results and conclusions. We've moved discussion of the methods/results of the pooled and panel data analyses to the SI.

Specific comments

Lines 278-282

Since, organic fields, while smaller in size, tend to be part of farms far larger than their conventional counterpart. Does that not influence the decision about what to spray and how much to spray compared to a farm which is exclusively organic. The exclusive organic farmers may be at a greater liberty to choose products with careful considerations in terms of both their environmental footprint and suitability for organic farming. I realise that there are certain products that are certified for organic farming.

In response to this comment and that of Reviewer 1, we went back to the pesticide use data to understand differences between purely organic and “mixed” farms (i.e. those that have both organic and conventional fields). In our data, less than 3% of organic fields in any year belong to exclusively organic farms. Rather, the majority are part of mixed farms. Given the very small sample size of exclusively organic fields, we did not pursue a more rigorous analysis. However, we note in lines 385-389 that our results may not apply to places with a greater preponderance of exclusively organic farms.

Line 249-252–

“A small number of observations ($n = 319$) were dropped due to missing soil quality data. Including observations with interpolated soil quality has little effect on our results (Figure S2).” Given this observation (soil quality not being a sensitive parameter) should these have been included in the analyses and would that have been beneficial in the second hurdle, such as in the case of self-reported organic (line 252)? Was this tested?

Thank you for the comment. We respectfully note that our full analysis sample is about 92,000 fields, of which 9000 are organic and 319 (total) are missing soil quality data. 319 represents 0.35% of our data. Nevertheless, we interpolated soil quality for those fields to test whether dropping them mattered. Unsurprisingly, given it was <1% of the sample, it didn't. Soil quality is an important covariate, is statistically significant in several models, and is systematically different between organic and conventional production (Table S1). We have removed Figure S2, because few would expect a change in outcomes with a reduction of the sample by 0.35% and it may be more confusing than clarifying.

Line 313-322...

In the absence of disaggregated yield data, availability of which “... would dramatically improve understanding of the environmental impacts of organic production” (line 321-322), how seriously could it undermine the current conclusions.

Yields are the missing link for evaluating the environmental impacts of various agronomic decisions. In this case, we found a fairly small difference in pesticide use and use for chemicals of potential hazard to various environmental outcomes without adjusting for yields. When we used global yield gap estimates, it shifted up the amount of pesticides used on organics, since they generally have lower yields. We note that organic fields are on intrinsically better soil in our study than their conventional neighbors so any estimated yield gap that didn't account for this would be an underestimate (where “intrinsic” is based on characteristics that are not readily influenced by management). However, this region is composed of intensive organic and conventional production so the yield gaps observed here could be smaller than the global average, were they produced on the same soil, etc.

In actuality, our main results that ignore yield differences are likely an overestimate of the difference between organic and conventional pesticide use, and the estimation accounting for yield gaps using data from a global meta-analysis is likely an

underestimate. Though these two approaches could be considered the upper and lower bounds of the difference between organic and conventional for pesticides, the story they tell is not altogether that different. Without adjusting for yield gaps, we see a non-significant coefficient on organics, as we do when we adjust for yield gaps. The point estimates shift up after adjusting for yield gaps, but the interpretation of no significant difference remains.

Rai Kookana

Reviewer #3 (Remarks to the Author):

The study represents a systematic comparison of pesticide use in organic and conventional horticulture in one county in California. It is based on sound econometric analysis and detailed field-level data, covering a large number of observations. The results are discussed in the light of the current literature and methodological strengths and weaknesses are explained. The topic has a high policy relevance and can inform the debate around sustainable intensification.

General remarks/questions:

It seems ok to first pool all observations and then run crop specific regressions. I would however make sure that the crop-specific estimation and the pesticide risk metrics are the main source for conclusions and policy recommendation.

Thank you for the comment. We have modified the discussion and conclusion to emphasize the crop-specific estimation and pesticide toxicity metrics.

The study make a considerable effort to address selection bias. The readers need to be made aware that it can also introduce aggregation bias if grapes and potatoes are pooled.

We agree. The aggregated data provides a useful overview, but the heterogeneity present in the crop-specific results illustrates that the overall average hides quite a bit of crop-specific heterogeneity in the relationship between organic status and pesticide use. We clarify that on lines 188-191.

To get an idea of the data behind each crop-metric specific estimation it would be very useful to have the number of observations indicated in the figures (which otherwise are very clear and informative).

The number of observations has been added to all legends in the main text. We note that the raw regression output tables are in the supplementary.

Also averaging the crop-specific results and comparing that to the pooled estimation results would be helpful.

We have added a comparison of the pooled results to the average of the crops included in the crop specific model (lines 202-205, SI Tables 5-6).

An unambiguous definition of pesticide use in organic farming as opposed to pesticide use in conventional farming is needed. What type of plant protection products are allowed for organic farming in the study site?

We have added additional clarification of “organic” in the methods (lines 426-435). In particular, we clarify that pesticide use in our study region is governed by the USDA’s National List of Allowed and Prohibited substances. The National List defines which synthetic pesticides are allowed and which natural pesticides are not allowed on organic fields, and their manner of use. There is not an invariable definition we can provide readers beyond describing the National List, which itself can be modified. In general, the majority of allowable products are non-synthetic biological, botanical and mineral inputs.

Why is there a need to make the study spatially explicit? It is not entirely clear why the pesticide use reports are not sufficient for performing the statistical analysis. Is this because there is no information about the certification status of a field in the pesticide use reports?

Yes. First, the pesticide use reports alone only provide data on fields that use pesticides. Not all fields use pesticides, and as our results suggest, organic fields are more likely to have no pesticide use. Second, there is no information on certification status in the pesticide use reports, or even whether the product is approved for use on organic agriculture.

It should be better explained how relevant the results are beyond the study region.

We have expanded discussion of the relevance beyond the study region on lines 382-391. In brief, we anticipate similar results in other intensive fruit and nut growing regions. However, areas with lower value crops or less intensive production are likely to observe different results than we do here.

Specific remarks/questions:

Rows 9-10: I agree that in terms of pesticide use there is a knowledge gap. However, I would not say that many of the environmental impacts of organic farming are poorly understood. There is a rich literature on comparative sustainability assessments. Per unit area, the evidence for many environmental impacts is quite clear. Per unit output there tends to be much more uncertainty. Figure 1 in this paper by Ramankutty et al. gives a good impression of the available evidence: <https://onlinelibrary.wiley.com/doi/abs/10.1111/agec.12534>.

Thank you for the comment. We have adjusted our wording to clarify we were talking about pesticides in particular (lines 8-9). We emphasize throughout that per unit output is

where the majority of knowledge gaps remain, which is why we tried, in a rudimentary way, to account for per unit output differences by accounting for average yield gaps. Additionally, we describe in the discussion, lines 392-397, that many environmental impacts are established and many are improved by organic practices, at least on a per area basis. In support of that statement, we cite the suggested paper.

Rows 22-23: "...but organic fields that are sprayed tend to receive similar levels of pesticides as their conventional neighbors." This can be misunderstood. The overall levels might be the same, but what does this information really mean? Active ingredients differ and pesticide risks should be the most policy-relevant metric here.

Thank you for the comment. We use both overall weight (active ingredients, products in kg/ha) as well as kg/ha of products that exceed regulatory thresholds for different environmental endpoints based on standard toxicity tests for several taxa. We explain these binary thresholds in the introduction (lines 76-78), methods (lines 512-519), results (lines 143-150), and note the caveats in the methods (615-624) and discussion (371-378). We slimmed the abstract to the required 150 words, which limits discussion of these important points in the abstract.

We tried to focus on an endpoint with substantial data, and at least compare the binary metrics of toxicity to a continuous metric for one endpoint. We chose fish, based on Nowell et al. 2014, who provided exceptional appendices for many chemicals in use in the US. We supplemented the Nowell et al. (2014) with data from Ecotox using the R package standtox (Scharmüller 2021). While we tried to follow the Nowell et al. (2014) methods, our mean toxicity concentrations differed for chemicals that existed in both the Nowell et al. (2014) data and our data. This was likely due to the treatment of outliers and various filters of studies included in the Nowell et al. (2014). As we are not toxicologists, we deferred to the Nowell et al. (2014) data where available, and used our estimates built with standtox where Nowell et al. (2014) were lacking data. Still, only about 70% of chemicals in use in our study period matched. Additionally, many products, which can include multiple chemicals, did not have full toxicity information. As the Pesticide Toxicity Index, as with other indices, is a weighted sum, those products without toxicity were not counted at all and products with chemicals missing toxicity were underestimated. Products used on organic fields were more likely to lack toxicity information for one or all chemicals, which biases our analysis. While chemicals lacking toxicity may be lower toxicity chemicals, we do not have the evidence or the expertise to support that claim.

Noting the bias due to non-random missing toxicity data, we nevertheless tried to run our double hurdle model for fields that had products with toxicity information for, on average, 70% of the chemicals. Just 35% of our organic fields that had pesticide use met this condition, while 67 % of the conventional fields with pesticide use met this condition. Our preliminary investigation suggests a decrease in toxicity weighted use (for fish) of about 50% for fields that are organic relative to conventional, depending on whether we adjust for yield gaps or not. Given the data limitations, we feel this is suggestive and worthy of future investigation, and we note the approach, highly

preliminary results and limitations in the methods (lines 615-624), results (lines 254-271) and discussion (lines 375-378).

Rows 31-34: A publication with more recent data (until 2019) is available. Data from the 2021 World of Organic Agriculture report puts the area under organic farming at 1.5% and 73.3 million ha. See the report here: <https://www.fibl.org/fileadmin/documents/shop/1150-organic-world-2021.pdf>

The citation and numbers have been updated as suggested.

Row 37: I suggest to mention the EU's Farm to Fork strategy here in addition to the paper from 2009. This is a recent paper on organic farming in the context of the F2F strategy: <https://onlinelibrary.wiley.com/doi/abs/10.1111/1746-692X.12294>

The reference has been added as suggested.

Rows 42-52: This is definitely true when studying actual behaviour. I would clarify that this is the intention here. It would still be important to also mention that there are long-term trials that compare organic and conventional cropping and that control for agro-ecological baseline conditions and management differences.

We have clarified we are discussing actual behavior and have added mention of long-term trials that control for baseline conditions (lines 35-38, lines 44-47).

Row 65: Which regulation are you referring to? There are multiple organic standards, which have different requirements. It would be good to check the COROS (Common Objectives and Requirements for Organic Standards) by IFOAM. In my understanding no chemical substances are allowed in organic farming if it is an officially recognised standard. Copper is a mineral substance not a chemical pesticide. Of course it is still hazardous.

We apologize for the lack of clarity. First, with regard to the regulations, we were referring to regulatory standards in the US, which is governed by a list of approved and prohibited substances. We clarified that in line 62, and explain the National List in lines 430-435. Because our study is focused in the US, we emphasize USDA organic certification. This certification can be considered organic in countries with trade agreements recognizing USDA organic certifications (www.ams.usda.gov/NOPInternationalAgreements). Examining the IFOAM COROS standards, particularly Chapter 3 Section B 4.5 and Appendix 3 (<https://www.ifoam.bio/sites/default/files/2020-09/IFOAM%20Norms%20July%202014%20Edits%202019.pdf>), the USDA standards often align with IFOAM organic certification, but IFOAM more specifically outlines allowed crop protectants. With regard to synthetic chemicals, while it is true that the vast majority of synthetic chemical substances are not allowed in (US) organic production, there are exceptions when there are no viable organic alternatives. Those

exceptions, at least in the US, are governed by the National List of Allowed and Prohibited Substances. Additionally, though we wouldn't observe it in our study, organic farmers are allowed to source plant propagation material from non-organic stock if no organic source is available commercially (US National Organics Program 205.204) or under certain regulatory (phytosanitary) requirements.

Row 67: I suggest to give more examples of a hazardous active ingredient in organic farming. Many know that copper is harmful, but it would be good for readers to be aware also of other active ingredients that are used in organic farming and that are (potentially) harmful to humans and the environment.

We've added mention of pyrethrin and azadirachtin (lines 64-66) as examples of active ingredients used in organic farming that are potentially harmful to the environment. We have rewritten this paragraph to make it clear that we are focusing on environmental impacts (lines 51-60).

Rows 89-94: Some more information on the sampling strategy would be helpful. From what population was the sample drawn? How were sample fields or farms selected? Or are all farms and fields in the county included?

Apologies for being unclear. We were trying to balance providing a sufficient taste of the methods so the reader could understand the results without replicating the methods section. We have added clarification that our sample was all fields permitted in Kern County, CA between 2013-2019 (lines 94-95), based on the Kern County Agricultural Commissioner's Office spatial data.

Rows 123-124: How did you control for these variables?

Thank you for the comment. We included covariates for soil quality (Storie Index), farm and field size, which we describe in greater detail in the methods. We've adjusted the wording to say including covariates for soil quality, farm and field size so it is no longer a distraction (line 137).

Rows 128-131: It is not fully clear that subsetting the data allows fully controlling for farm and farmer characteristics? Are there for example farms that operate different enterprises for organic and conventional vegetable production or that have different managers depending on the production system? Also, by reducing your sample to farms with both types of fields your results are no longer representative for the larger population of horticultural farms in the region. It would be interesting to perform the analysis also using the entire dataset and relying on reweighting methods to control for selection bias. This should of course not substitute the present analysis, but it can be an interesting robustness check. You mention that the use of propensity score matching does not lead to sufficient balance. Entropy weights are an appealing alternative, because of the exact adjustment of covariate moments. For more information see <https://web.stanford.edu/~jhain/Paper/PA2012.pdf>.

We agree subsetting is subideal for all of the reasons you mention-- it results in a loss of $\frac{2}{3}$ of our data, there very likely are farms that have different enterprises though we do not observe that structure, and the subset is unlikely to be representative. Both subsetting and panel methods were our initial starting point. We used them to contrast the results with the double hurdle model, which models the true zero use decision. Because of the confusion noted here and the request by Reviewer 2 to reduce the technical aspects of the methods, we decided to relocate the pooled OLS and panel data methods and results to the SI and just jump into the log-normal hurdle model.

Row 132: You wrote earlier that the number of organic fields ranged from a low of 936 in 2013 to a high of 1,544 in 2017. How did you account for conversion to organic or abandoning certification in your panel data model?

We include year effects to absorb shocks to production methods or prices, but we don't track fields entering or exiting. If the reviewer thinks tracking these fields is valuable, it can be done. The only potential way we see bias stemming from not tracking is through modified methods on fields that will eventually become organic. However, those represent at most a couple hundred out of ~12,000 fields per year. Additionally, since organic takes several (3) years of modified management practices, we would need to throw out the last 2 years of our sample since we do not know which fields will end up being organic in 2020 (which is outside of our data). We believe that would cause more bias than the contamination caused by the few fields that are not labeled organic but are transitioning. If the reviewer disagrees or has other ideas for how to proceed, we are happy to address further comments on this point.

Rows 152-155: A hurdle model seems a plausible choice here. It would be good to explain that this model choice implies that zeros represent the true choice of the farmers rather than a decision that the farmer had no control over for some reason.

Thank you for the suggestion. We have added a clarification on this point, lines 134-136.

Rows 161-170: It is definitely true that overall pesticide use quantity is not a good indicator of how hazardous an active ingredient is for the environment and humans. It needs to be justified why the EPA pesticide label information is good enough to quantify harm. There are a large number of metrics available that measure this, e.g. the Environmental Impact Quotient, the Pesticide Load Index or the Environmental Yardstick. The following paper by Möhring et al. illustrates the relevance of pesticide indicator choice for the results: <https://onlinelibrary.wiley.com/doi/full/10.1111/agec.12563>. If it is difficult to calculate a more refined index because of the diversity of active ingredients used, it would be interesting to know how robust the EPA information is against these more advanced metrics for the 10 most important active ingredients in the region.

Thank you for this important note. We agree. As noted in our response to the comments of prior reviewers, we tried to address the lack of toxicity critiques by adding the calculation of the Pesticide Toxicity Index (Nowell et al. 2014). We chose the Pesticide Toxicity Index rather than other metrics because of its coverage of chemicals in use in US agriculture. Still, not all of our chemicals matched and chemicals used on organic fields were much more likely to be missing toxicity information. As the Pesticide Toxicity Index, like other indices, is a weighted average, missing toxicity information results in an underestimate of toxicity-weighted pesticide use.

We include ecotoxicological data for products representing about 50% of use by weight for organic and conventional fields in table S10. These products are defined by EPA registration number. We considered providing Environmental Impact Quotient values, as the reviewer suggested, for these commonly used products. Based on the May 2020 version of the EIQ values table obtained from nspim.cornell.edu (here), there are no EPA registration numbers provided or CAS numbers for the active ingredients. Using a rudimentary match on either the “common name” or “trade name” columns in the EIQ table, again results in numerous missing values even for our most commonly used products (e.g. Sonata, Suppress EC) and active ingredients (e.g. capric or caprylic acid, various Bacillus strains, mineral oil etc).

We now include additional clarification on the determination of potential hazards on the environmental hazards statement of the pesticide label (lines 512-519, 143-150), and note the caveats in the methods (615-624) and discussion (372-378). These statements are based on standard toxicity tests for birds, fish, invertebrates, pollinating insects (bees), and mammals with consistent thresholds for labeling decisions. For example, the EPA typically requires that a label indicate that a product is toxic to birds if any active ingredients has an avian acute oral LD50 value less than or equal to 100 mg/kg, toxic to fish or invertebrates if the fish acute LC50 or aquatic invertebrates EC50 value is less than or equal to 1 ppm, and toxic to mammals if the mammalian acute oral LD50 value is less than equal to than 100 mg/kg. Toxicity labeling for pollinating insects is broken into two categories—products must be labeled as highly toxic to pollinating insects if any active ingredient’s acute LD50 is less than 2 micrograms per bee or moderately toxic if the acute LD50 value is between 2 and 11 micrograms per bee.

Figure 2: Why is the change in terms of high toxicity product use not reflected in the bee and fish hazard indicators? Does high toxicity only refer to humans in this case?

We have added clarification to the legend that high toxicity only refers to acute toxicity to people. We are basing “high” and “low” on the EPA signal words, which are for acute toxicity to people.

Rows 227-228: This is an interesting finding. What drives this massive increase in use?

We parsed pesticide use between organic and conventional grapes in greater detail. We looked at products with dual action (insecticides/fungicides) as well as single action

(insecticides). Organic grapes use substantially more products functioning as fungicides, and in particular those functioning as insecticides/fungicides. These are chemicals like sulfur products, some of which are organic approved (others aren't due to adjuvants). Organic grapes also used more products that functioned as herbicides, particularly more products that functioned as only herbicides or herbicides and fungicides.

Row 276: What does intrinsically better soil refer to? Soil type?

With "intrinsic" we were trying to indicate soil characteristics that are not easily modifiable by the farmer. The Revised Storie Index is composed of several factors, some of which are potentially modifiable by the farmer (e.g. pH) and others of which are not (e.g. landscape scale slope angle). The "dynamic" or more modifiable characteristics are in the "Factor X" aka "dynamic" properties. We show in table S1, that organic fields have a lower Storie Index rating (where lower indicates better soil), after controlling for crop type. Further, we show that this difference is not driven by the dynamic properties since there is very little difference, on average, between organic and conventional fields in the main components of Factor X (again after controlling for crop type).

The Revised Storie Index rating system relies on digital soil survey data stored in the National Soil Information System (NASIS). Input data used to determine Storie Index ratings represent landscape-scale factors such as landform and topography, as well as chemical properties such as pH and electrical conductivity. Data are acquired from NRCS National Cooperative Soil Survey and are publicly available in the SSURGO II database. Soil factors which are easily modifiable such as fertility and microtopography are excluded from the Revised Storie Index Rating, yet there remain some factors that are more easily modifiable than others and those are delegated to "Factor-X". Intrinsically better soil thus refers to locations where physical properties such as slope angle and soil depth, and chemical properties such as pH and electrical conductivity are more favorable for agricultural production, based on the Revised Storie Index Rating, and these are not driven by dynamic properties more easily modifiable by the farmer.

Rows 308-309: This shows that recommendations primarily need to be crop-specific. In addition, pesticide risk rather than use levels for individual crops would be the most valuable information for designing capacity development interventions and policies.

We agree. We now emphasize the importance of crop-specific results and policy in the discussion.

Rows 361: Could you provide some examples where you expect similar production conditions and production systems than in Kern County. This would be helpful to get a better idea in how far the results also apply to other regions.

Thank you. We have added lines 382-391 to describe how these results apply to other regions. In brief, we anticipate similar results in other intensive fruit and nut growing

regions. However, areas with lower value crops or less intensive production are likely to observe different results than we do here.

Citations

Farrar, J. J., Baur, M. E., & Elliott, S. F. Adoption of IPM practices in grape, tree fruit and nut production in the western United States. *Journal of Integrated Pest Management* 7, 1–8, (2016).

Nowell, L. H., Norman, J. E., Moran, P. W., Martin, J. D. & Stone, W. W. Pesticide Toxicity Index—A tool for assessing potential toxicity of pesticide mixtures to freshwater aquatic organisms. *Science of The Total Environment* 476–477, 144–157 (2014).

Scharmüller, A. Package 'standartox'. Ecotoxicological information from the Standartox Database. CRAN (2021).

Reviewers' Comments:

Reviewer #1:

Remarks to the Author:

I think the authors have done a nice job of responding to my and other reviewers in their response to review. The manuscript reads very well! I have no further comments.

Reviewer #2:

Remarks to the Author:

The authors have taken the detailed comments, previously made by the two other two reviewers and I, into account and have revised the manuscript substantially.

I noted that the comments of the three reviewers had a general congruence in terms of some of the weaknesses in the previous manuscript. However, the authors have responded comprehensively to these and in response revised the manuscript substantially.

I, for one, am satisfied with the revised version of the manuscript and happy to recommend its publication.

Reviewer #3:

Remarks to the Author:

Thank you very much for your detailed and thorough replies to the comments and the related revisions in the text. From my perspective, all concerns have been addressed and the manuscript is excellent now. It does not need further revisions.

RESPONSE TO REVIEWER COMMENTS

Please find the reviewers' comments, produced verbatim, and our responses indented in italics.

We acknowledge the extremely helpful comments of all 3 reviewers on the initial submission and appreciate their time and energy reviewing our revised version.

REVIEWERS' COMMENTS

Reviewer #1 (Remarks to the Author):

I think the authors have done a nice job of responding to my and other reviewers in their response to review. The manuscript reads very well! I have no further comments.

Thank you.

Reviewer #2 (Remarks to the Author):

The authors have taken the detailed comments, previously made by the two other two reviewers and I, into account and have revised the manuscript substantially.

I noted that the comments of the three reviewers had a general congruence in terms of some of the weaknesses in the previous manuscript. However, the authors have responded comprehensively to these and in response revised the manuscript substantially.

I, for one, am satisfied with the revised version of the manuscript and happy to recommend its publication.

Thank you.

Reviewer #3 (Remarks to the Author):

Thank you very much for your detailed and thorough replies to the comments and the related revisions in the text. From my perspective, all concerns have been addressed and the manuscript is excellent now. It does not need further revisions.

Thank you.